# Duodenases are a small subfamily of ruminant intestinal serine proteases that have undergone a remarkable diversification in cleavage specificity

Zhirong Fu[1], Srinivas Akula[1], Chang Qiao[1], Jinhye Ryu[1], Gurdeep Chahal[1], Lawrence de Garavilla[2], Jukka Kervinen[3], Michael Thorpe[1], Lars Hellman[1]*

1 Department of Cell and Molecular Biology, The Biomedical Centre, Uppsala University, Uppsala, Sweden,
2 GDL Pharmaceutical Consulting and Contracting, Downingtown, Pennsylvania, United States of America,
3 Tosoh Bioscience LLC, King of Prussia, Pennsylvania, United States of America

* Lars.Hellman@icm.uu.se

**Data Availability Statement:** All relevant data are within the manuscript and its Supporting information files.

## Abstract

Ruminants have a very complex digestive system adapted for the digestion of cellulose rich food. Gene duplications have been central in the process of adapting their digestive system for this complex food source. One of the new loci involved in food digestion is the lysozyme c locus where cows have ten active such genes compared to a single gene in humans and where four of the bovine copies are expressed in the abomasum, the real stomach. The second locus that has become part of the ruminant digestive system is the chymase locus. The chymase locus encodes several of the major hematopoietic granule proteases. In ruminants, genes within the chymase locus have duplicated and some of them are expressed in the duodenum and are therefore called duodenases. To obtain information on their specificities and functions we produced six recombinant proteolytically active duodenases (three from cows, two from sheep and one from pigs). Two of the sheep duodenases were found to be highly specific tryptases and one of the bovine duodenases was a highly specific asp-ase. The remaining two bovine duodenases were dual enzymes with potent tryptase and chymase activities. In contrast, the pig enzyme was a chymase with no tryptase or asp-ase activity. These results point to a remarkable flexibility in both the primary and extended specificities within a single chromosomal locus that most likely has originated from one or a few genes by several rounds of local gene duplications. Interestingly, using the consensus cleavage site for the bovine asp-ase to screen the entire bovine proteome, it revealed Mucin-5B as one of the potential targets. Using the same strategy for one of the sheep tryptases, this enzyme was found to have potential cleavage sites in two chemokine receptors, CCR3 and 7, suggesting a role for this enzyme to suppress intestinal inflammation.

**Funding:** GDL Pharmaceutical Consulting and Contracting provided support for this study in the form of salary for LG and Tosoh Bioscience LLC provided support for this study in the form of salary for JK. The specific roles of these authors are articulated in the 'author contributions' section. Knut och Alice Wallenbergs Stiftelse provided support to LH (KAW2017.0022). The funders had no role in study design, data collection and analysis, decision to publish, or preparation of the manuscript.

**Competing interests:** The authors have read the journal's policy and the authors of this manuscript have the following competing interests: LG is a paid employee of GDL Pharmaceutical Consulting and Contracting and JK is a paid employee of Tosoh Bioscience LLC. There are no patents, products in development or marketing products to declare. This does not alter our adherence to PLOS ONE policies on sharing data and materials.

**Abbreviations:** EK, enterokinase; Ni-NTA, nickel-nitrilotriacetic acid.

## Introduction

Ruminants have a very complex digestive system with several separate compartments and with the possibility to ruminate. Their complex digestive system makes it possible for them, with the help of bacteria, protozoa and fungi, to utilize cellulose as an energy source. To enhance the efficiency of their digestive system, several interesting additions to the enzyme repertoire have occurred by gene duplications. Two important such increases in the digestive repertoire come from two loci originally involved in immunity. One is the lysozyme c gene, the conventional type of lysozyme or chicken lysozyme and the second is the serine proteases encoded from the mast cell chymase locus. In both of these loci, the gene number has increased and the new genes have changed tissue specificity from primarily being expressed in immune cells to now being expressed in the intestinal region of these animals.

In humans we have one lysozyme c gene that is primarily expressed in macrophages and neutrophils. In mice there are two, the M and P lysozyme genes, where the first, similar to the human gene, is primarily expressed by immune cells whereas the second, the P gene, is instead expressed by epithelial cells in the intestinal region and the lung [1]. In sharp contrast to both mice and humans, cows have ten individual lysozyme c genes, and sheep has seven [2]. A number of them are expressed in the intestinal region and there most likely taking part in food digestion (Fig 1). In cows, at least four of these lysozyme genes are expressed in the intestinal region [3]. There, the lysozyme transcription level is also extremely high and has been shown to account for 10% of the total transcriptome in the intestinal region [4]. In contrast to these high levels in the intestinal region, cows have been found to have very low levels of lysozyme in the milk, only around 10 μg/l or less whereas human breast milk contains 10 000 times higher lysozyme levels, in the range of 100 mg/l [5,6]. Relatively low levels of lysozyme have also been detected in the blood of cows, a factor that may have effects on their immunity to bacterial infections and to the development of mastitis [7]. Interestingly, horses have even higher lysozyme levels than humans in their milk, close to ten times, 0.8–1.3 g/l, showing the large differences between different placental mammals in the presence of antibacterial proteins in different organs and secretions [8].

A similar expansion in the number of genes as for lysozyme c has occurred in another locus, the mast cell chymase locus. This is a locus that in most mammals encodes a few trypsin/chymotrypsin-related serine proteases expressed by hematopoietic cells. In both sheep and cows, a number of gene duplications have resulted in several new protease genes compared to the human locus. There are thirteen new genes in sheep, ten in cows and two in pigs (Fig 2). Four of these new bovine genes and five of the sheep proteases belong to a new subfamily of proteases that are expressed in the intestinal region and therefore have been named duodenases (marked in red in Fig 2). A slightly more distantly related gene has also appeared in the pig chymase locus (Fig 2). In contrast to the lysozyme and lysozyme-related gene family, very little is known about these new ruminant and pig intestinal proteases. These proteases belong to the large family of trypsin-chymotrypsin-type serine proteases that, in mammals, are involved in a diverse array of biological functions such as blood coagulation, complement activation, food digestion, fertilization, apoptosis induction, cytokine inactivation, connective tissue turnover and general inflammation. This serine protease family is one of the major protease families in mammals. A number of them are stored in cytoplasmic granules of hematopoietic cells, i.e., mast cells, neutrophils, cytotoxic T cells and NK cells [9–12]. In mammals, these hematopoietic serine proteases are encoded from four different loci, the chymase locus, the met-ase locus, the T cell tryptase and the mast cell tryptase loci [13]. In cattle, sheep and pigs, genes within the chymase locus have duplicated and a few of the new copies are expressed at high levels in the intestinal region, hence the name duodenases [13,14]. The cow

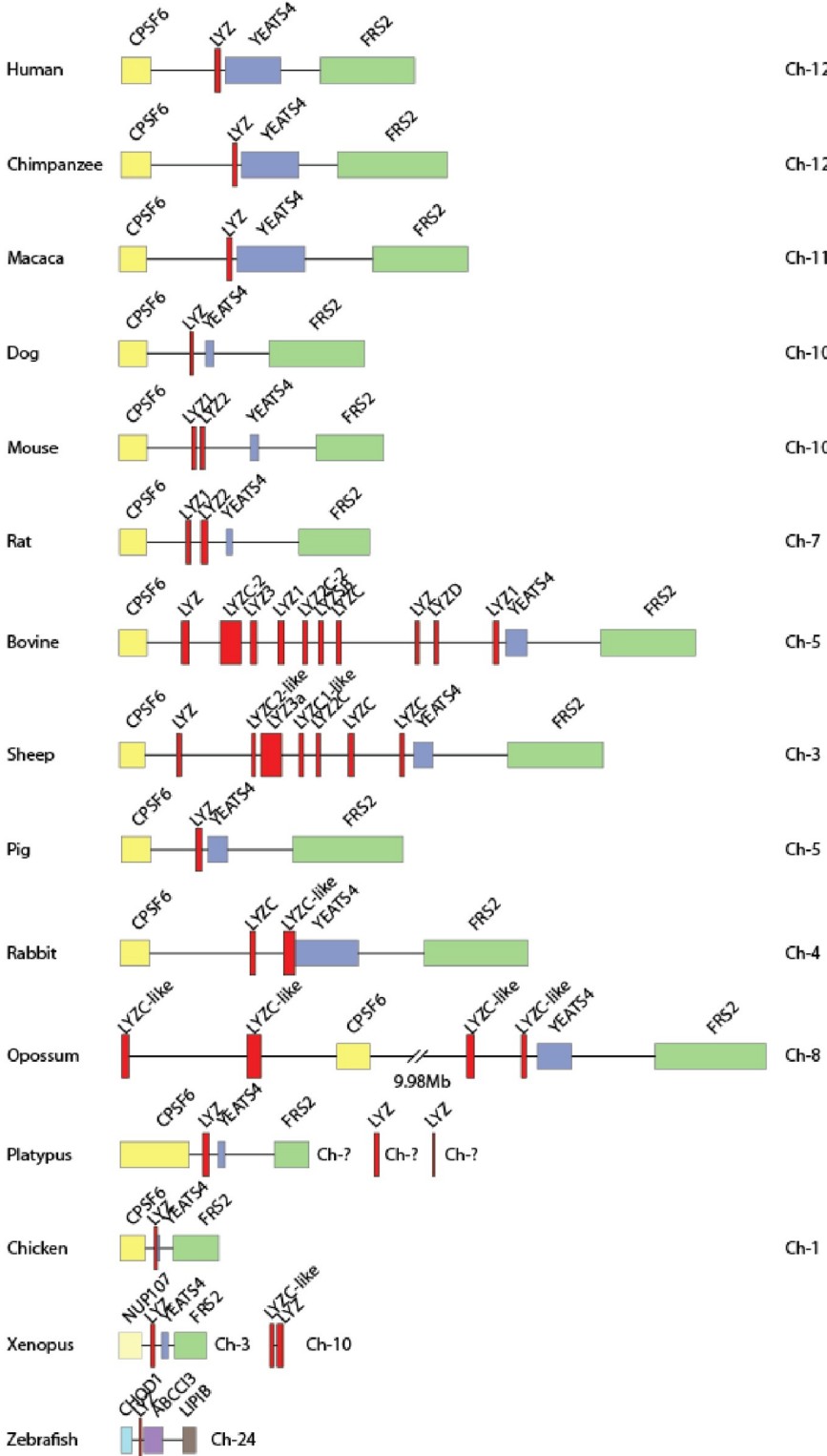

**Fig 1. The lysozyme c locus.** The lysozyme c locus with flanking genes are shown in scale in a panel of different vertebrate species from fish to mammals to highlight the massive expansion of lysozyme c gene that has occurred in ruminants as here represented by sheep and cow.

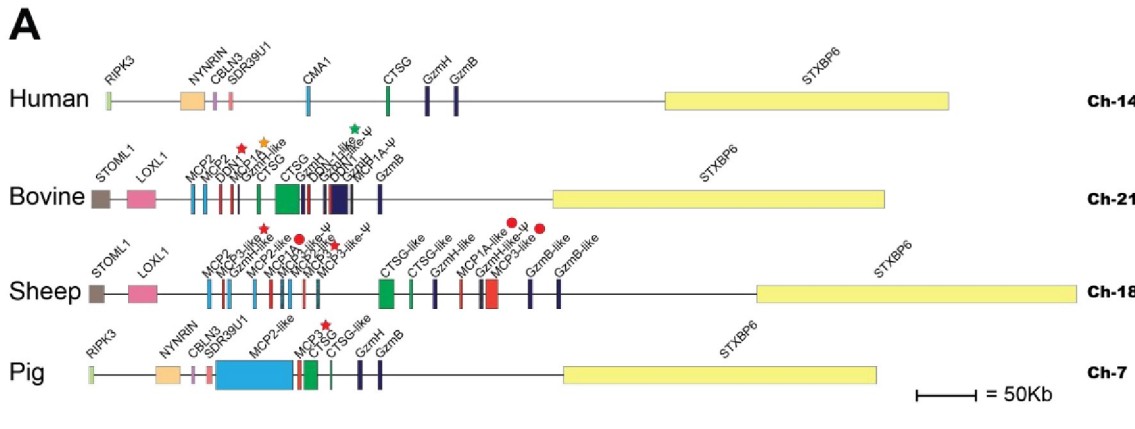

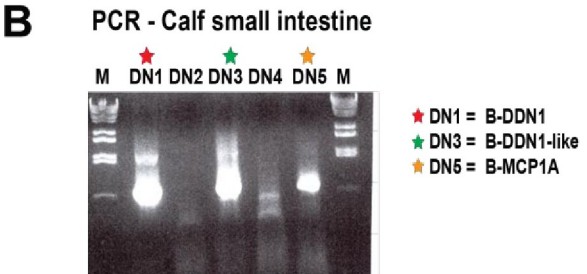

**Fig 2. The chymase locus and a PCR analysis of the expression of duodenases in cow duodenum.** The chymase locus of four mammalian species, human, bovine, sheep and pig is presented in panel A. This is an in-scale presentation of this locus including a few flanking non-lysozyme genes. Panel B shows a PCR analysis of the originally five duodenases in the early version of the cow chymase locus. As can be seen from the figure, only three of the genes show a detectable PCR band of the correct size. In a recent genome update it has been shown that these three genes are complete and most likely functional and the fourth is a pseudogene. This locus does now in the most recent genome update only contain four duodenase genes, three active and one pseudogene. The most recent update of the sheep locus now contains five duodenase genes where four seems to be functional. The second MCP3-like marked with a red dot is most likely non-functional. The genes we have characterized are marked with stars and the ones not yet analysed by red dots. The stars for the bovine enzymes are colour coded as for panel B, the PCR analysis.

duodenases have been shown by both immunofluorescens and immune gold staining to be expressed primarily in duodenal intestinal epithelial Brunner's glands, and no expression was observed in infiltrating hematopoietic cells of that tissue [15]. A duodenase has also been isolated from parasite infested sheep duodenal tissue [16]. However there no analysis of what cell type that expressed this protease was performed [16]. Two duodenases were later cloned from sheep bone marrow derived mast cells (BMMCs) and abomasum tissue named sheep mast cell proteases 1 and 3, s-MCP-1 and sMCP-3 [17]. Based on the expression of the cow duodenases in the Brunner's glands and their secretion into the duodenal lumen, these new proteases are thought to be involved in food digestion or possibly in the activation of other food-digesting enzymes, similar to enterokinase that activates pro-trypsin into active trypsin [15,18]. The cow duodenases have actually been shown to be able to activate recombinant proenteropeptidase, the enterokinase, that in turn activate trypsin [18].

However, as described above contradicting results concerning tissue expression has come from two later studies where cells with similar distribution and numbers as mast cells in bovine intestinal samples have been shown to stain with a polyclonal duodenase serum [19]. Using an antiserum against the sheep duodenase in the same study an analysis of worm infected bovine lung indicating that at least during inflammatory conditions the duodenases can be expressed also by hematopoietic cells, most likely mast cells in the lung [19]. To this should also be added

the above described cloning of sheep MCP-1 and sheep MCP-3 from RNA of BMMCs and abomasum [17]. Although the majority of the duodenases expressed in cow duodenum seems to originate from the Brunner´s glands and secreted into the intestinal lumen, there are still questions concerning the expression also in mucosal mast cells in both sheep and cow and if these proteases have both digestive and immunomodulatory functions [20].

Based on a phylogenetic analysis of chymase locus-encoded genes, the duodenases seem to be more closely related to the T and NK cell granzymes and neutrophil cathepsin G than to the mast cell chymases (Fig 3) [13]. The primary and extended specificities of the majority of the chymase locus-encoded genes have been determined. The human mast cell chymase, cathepsin G and granzyme H, all primarily show chymotryptic activity [21–23]. However, human cathepsin G also cleaves after lysine, and therefore shows tryptase-type activity as well [22,24]. Granzyme B is an asp-ase and accordingly cleaves after negatively charged amino acids such as aspartic acid [25,26]. However, as to the cleavage specificity, few studies are available for duo-denases. Initial tests with chromogenic substrates have shown that a preparation of two cattle duodenase isoforms cleaves substrates containing both an arginine or a phenylalanine in the P1 position and thereby have both tryptic and chymotryptic activities [14]. The duodenase iso-lated from sheep duodenal tissue did also show dual tryptase and chymase activity [16]. These proteases have also been shown to represent the dominant endopeptidase activity of the bovine duodenal mucosa and constitute 0.3–0.4% of the total protein of a duodenal mucosal homoge-nate [14]. However, little is known about their potential targets and their role in ruminant and pig digestive system. As the first step in resolving this and other questions concerning this new subfamily of proteases, we here present the extended cleavage specificities for six of these enzymes, three bovine, two sheep and one pig enzyme. Our results show that these enzymes, following the gene duplications, have diversified extensively in cleavage specificity. These new members of the chymase locus have obtained cleavage specificities as different as primarily tryptase activity, primarily chymase activity, enzymes with dual tryptase and chymase activities and one with preference for negatively charged amino acids. Some of these new enzymes have a relatively high selectivity indicating a few selected substrates and thereby a regulatory func-tion whereas others have a broad specificity and may in contrast have more general digestive functions similar to the pancreatic enzymes, trypsin, chymotrypsin and pancreatic elastase. We also used the consensus sequences obtained from the cleavage specificity analysis to iden-tify potential in vivo targets, and obtained a few interesting substrate candidates including sali-vary Mucin-5B and several chemokine receptors.

## Results

### Production, activation and purification of recombinant sheep, cow and pig duodenases

To ensure that the enzymes are pure and not a mix of very closely related enzymes with poten-tially different primary and extended specificities, all the six proteases used here were produced as recombinant enzymes in insect or mammalian cells (Fig 3). The coding regions for two sheep serine proteases, that had previously been isolated as cDNAs from sheep bone marrow mast cell cultures and intestinal tissue samples, respectively, and named mast cell proteases II and III (MCP2 and MCP3), were inserted in the baculovirus vector pAcGP67B and expressed in baculovirus-infected insect cells as previously described [27]. MCP3 clusters with the duo-denases and MCP2 with the mast cell chymases (Fig 3). Following purification, the fully mature MCP3 enzyme was ≥95% pure as determined by SDS-PAGE (Fig 4) and the correct N-terminus was confirmed by N-terminal sequencing. Mass spectral analysis indicated that sheep MCP3 is heterogeneously glycosylated which is also noticeable in SDS-PAGE as a fuzzy

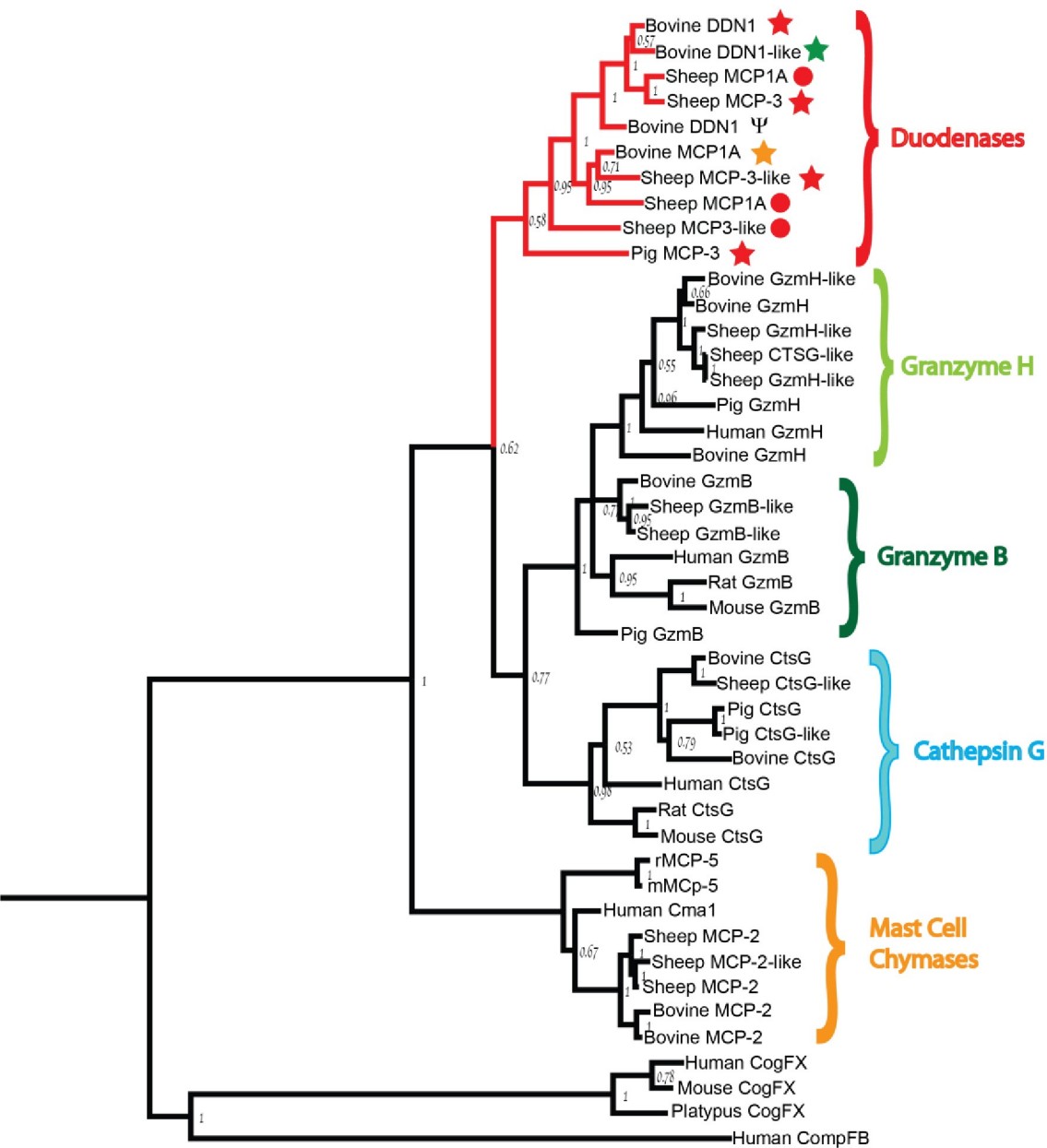

**Fig 3. A phylogenetic analysis in the form of a phylogenetic tree of chymase locus genes.** A number of mammalian chymase locus genes have been analysed for sequence relatedness using the neighbour joining algorithm. Human complement factor B and a few coagulation factor X proteins were used as outgroup. Duodenases form a clear subfamily distinct from the cathepsin G, granzymes and chymases. The six genes analysed in more detail for their cleavage specificity is marked by stars as in the Fig 2 and the genes not yet characterized are marked with a red dot, also similar to the Fig 2.

band of 25–30 kDa (Fig 4) [27]. Another sheep chymase (MCP2, not characterized here) is a classical mast cell alpha-chymase and closely related to the human mast cell chymase. This enzyme has been analysed previously and found to have a chymotryptic specificity similar to previously studied classical mast cell chymases [28].

To broaden the analysis of the duodenases, we decided to also determine the specificity of three bovine and one porcine duodenase. The initial sequence information for the bovine

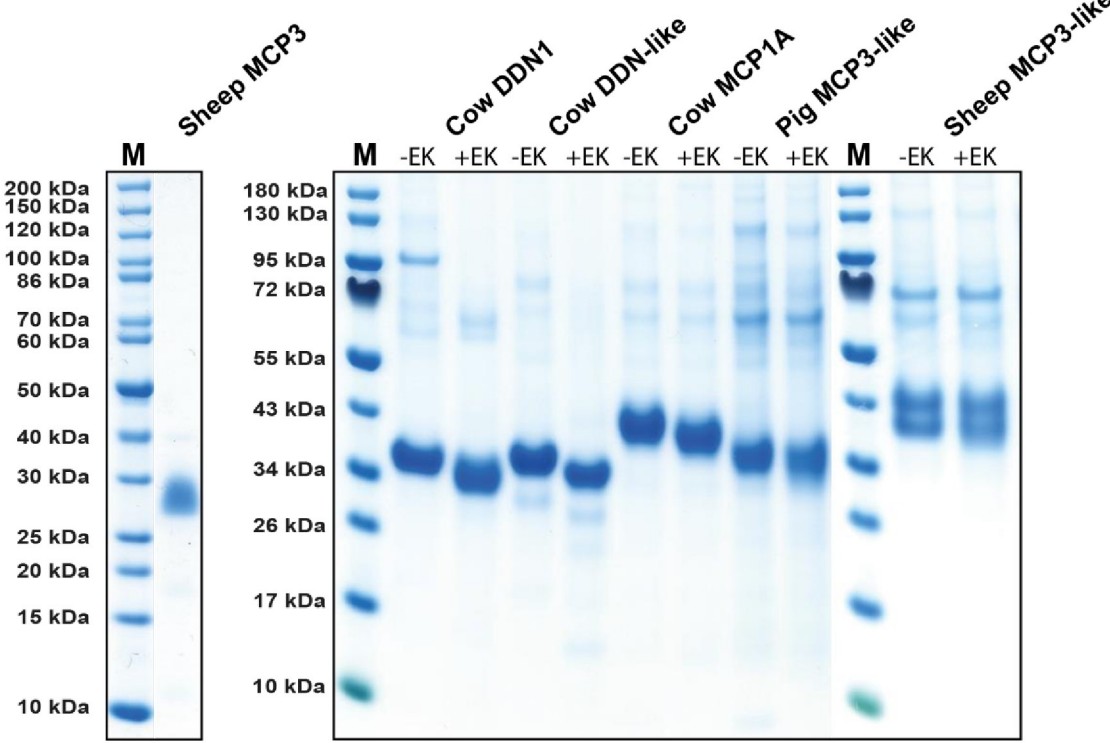

**Fig 4. SDS-PAGE of the recombinant sheep, cow and pig duodenases.** The recombinant sheep MCP3 was expressed in a baculovirus expression system. These proenzymes were first purified on Ni-NTA beads (-EK) and then activated by removal of the His$_6$-tag by enterokinase digestion (+EK). Following activation, the enzyme was further purified over heparin-Sepharose columns. In the right part of the figure we show the recombinant cow, pig and sheep duodenases, cow DDN1, cow DDN-like, cow MCP1A, Pig MCP3-like and the second sheep duodenase sheep MCP3-like. All these proteases were produced in the human cell line HEK293-EBNA using the episomal vector pCEP-Pu2. After purification on IMAC columns they were activated by enterokinase cleavage. The proteins are shown before and after enterokinase cleavage. Enterokinase removes the N terminal six his tag and the enterokinase site resulting in a reduction in size of the protein by approximately 1.5 kDa. The enzymes were after purification analysed by separation on SDS-PAGE and visualized with Coomassie Brilliant Blue staining.

genome had indicated five duodenase genes. Primers were prepared for all of them and PCR analysis performed of calf intestinal cDNA samples (small intestine). Only three of the primer sets resulted in clear and strong bands of the expected size (Fig 2B). From the now updated cow genome sequence, we can see that the three primer pairs for which we got good signals, DN1, DN3 and DN5, they correspond to the three of the four bovine duodenase genes that are present in the updated genome assembly, the genes DDN1, DDN1L and MCP1A (Fig 2). The successful isolation of three of the presently four duodenases that are present in the current version of the cow genome also shows that they are actively transcribed in the duodenum of the cow. Based on the genome sequence of the fourth gene, it appears to be a pseudo gene (Fig 2). The strong PCR bands from the calf intestinal sample also indicate that all three ¨functional genes¨ are expressed at relatively high expression levels in the intestinal region (Fig 2B). These three bovine cDNA sequences, and one of the two almost identical genes for the pig duodenase, were ordered as designer genes from Genscript and inserted (individually) in the mammalian expression vector pCEP-Pu2. All four expression plasmids were transfected individually into the human embryonic kidney cell line HEK293-EBNA for expression. After establishing confluent cultures of the transfected cells, the medium was collected and the secreted recombinant proteins were purified on Ni-chelating IMAC column. After purification, all three cow duodenases and the pig duodenase yielded sufficient amounts of product

for characterization (Fig 4). Following purification, the enzymes were activated by removing the N-terminal His-6 tag and the enterokinase site by cleavage with enterokinase. The proteins before and after enterokinase cleavage are shown in the Fig 4. We also cloned and expressed an additional active sheep duodenase originating from the MCP3-like gene. This duodenase was also expressed in the HEK293 cells using the pCEP-Pu2 vector (Fig 4).

In the most updated version of the sheep chymase locus, the number of genes has changed and there are now five duodenase genes, where all seems active (Figs 2 and 3). The number of active duodenases in cow, sheep and pig are now nine of which we have produced recombinant protein for six of them.

## Chromogenic substrate assay for the recombinant sheep and cow duodenases

A panel of chromogenic substrates were tested to study the primary specificity of the total of six recombinant duodenases from sheep, cow and pig. Interestingly, one of the sheep duodenases (MCP3) showed cleavage of both tryptase and chymase substrates, however, to a lesser extent of the chymase substrates (Fig 5). Two of the cow duodenases DDN-1and DDN1-like enzymes showed both strong chymase and a slightly lower tryptase activity whereas the third bovine enzyme MCP1A was found to be an asp-ase, only cleaving substrates with a negatively charged amino acid in the P1 position (Fig 5, substrates "J" and "K"). The pig enzyme cleaved only chymase substrates with strong activity, much higher on a molar basis than the sheep enzyme, indicating that it is a pure chymotryptic enzyme (Fig 5). For the second sheep duodenase, the MCP3-like, we were not able to detect cleavage activity of any of the chromogenic substrates using the same panel of substrates as for the other duodenases. Human mast cell chymase and human thrombin were included as reference proteases for chymase and tryptase activities (Fig 5) [21,29,30].

## Determination of the extended cleavage specificity by phage display

We performed a detailed analysis of the extended specificity by using phage display technology for four of the above enzymes, the two sheep duodenases, the cow asp-ase and the pig duodenase. The reason for selecting these four enzymes for phage display was based on the fact that three of them represented three different primary specificities, tryptase, asp-ase and chymase, and one of them was not possible to obtain any information concerning its primary specificity based on the fact that it did not cleave any of the chromogenic substrates. For the two remaining duodenases that showed both chymase and tryptase activity, phage display has a tendency to only select the most preferred specificity and overlook a second activity if it is slightly less preferred, which has been seen for human cathepsin G, which also has dual activity being both a chymase and a tryptase with a tryptase specificity for lysine [22].

For the phage display we used a library where each phage clone expresses a unique sequence of 9 random amino acids, followed by a His$_6$-tag in the C-terminus of capsid protein 10. Thereby the phages display a random nonamer on their surface and by interactions of the His$_6$-tag the phages can be immobilized on Ni-NTA agarose beads. Generally, only one copy of this modified capsid protein 10 is present on each phage. The phage library we used contains approximately $5 \times 10^7$ such phage clones.

The first enzyme to be analysed by phage display was the recombinant sheep enzyme MCP3. After the first selection step (biopanning), the phages, released by digestion of nonapeptides, were amplified in *E. coli* and subjected to additional biopannings. Selections of nonamers, susceptible to cleavage by the protease, was performed over five biopannings, after which it induced the release of more than 200 times more phages, compared to a PBS control.

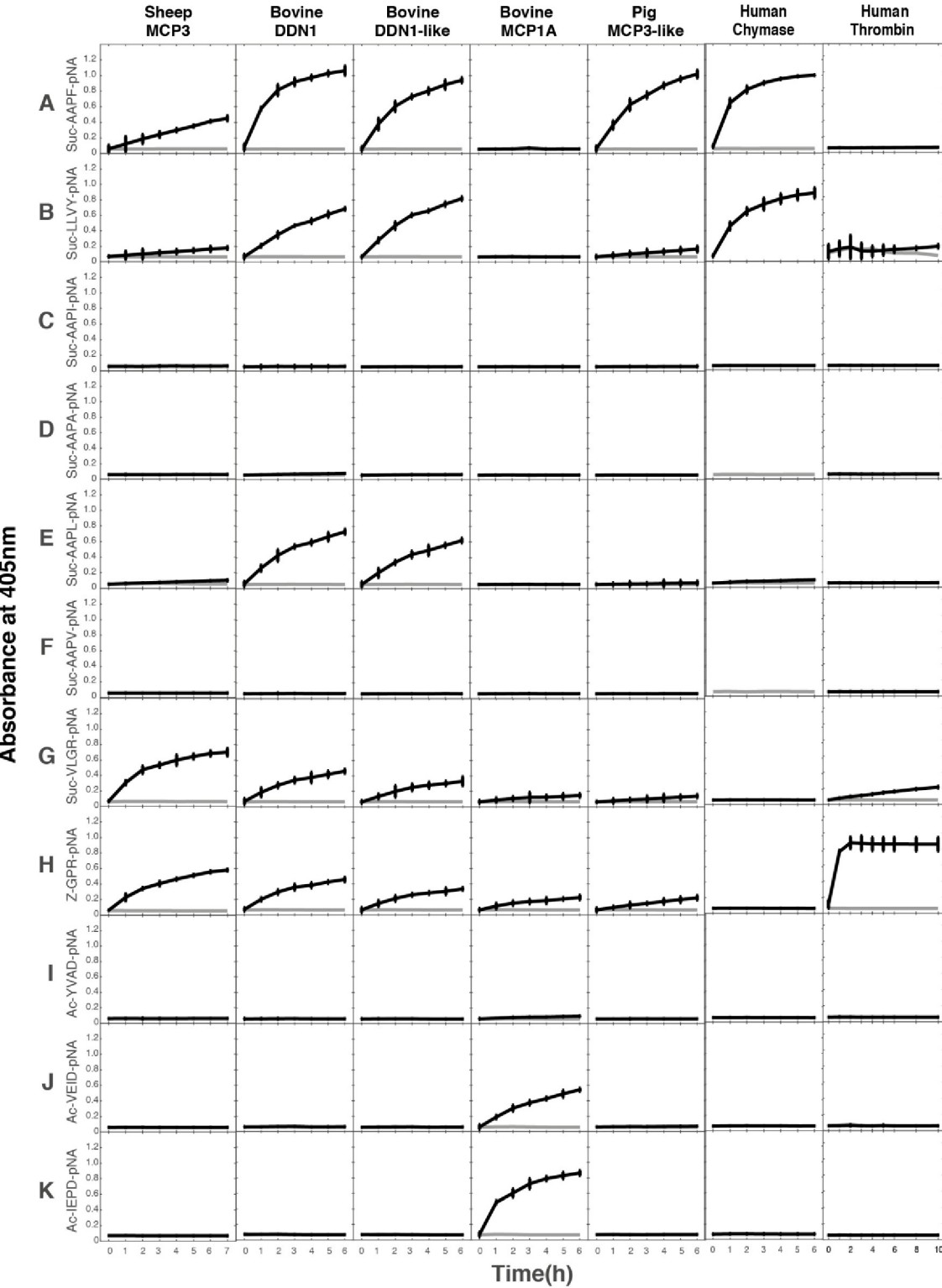

**Fig 5. Chromogenic substrate analysis of the sheep, cow and pig duodenases.** A panel of chromogenic substrates was used to determine the primary specificity of the different duodenases. Specificities of the human chymase and human thrombin are presented as reference. The panel includes different chymase, elastase, tryptase and asp-ase substrates. The amino acid sequences of the substrates are listed at the left side of the panels. The analyses were done in triplicates and the standard deviation bars within these triplicates is presented in the figure.

After the last biopanning, 75 individual phage clones were isolated. After PCR amplification of the region of interest, 60 PCR fragments were sent for sequencing. Of these 60 clones, 57 gave readable sequence. The sequences of these PCR fragments contained the sequence encoding the randomly synthesized nona-peptides was decoded and aligned (Fig 6). Upon comparative

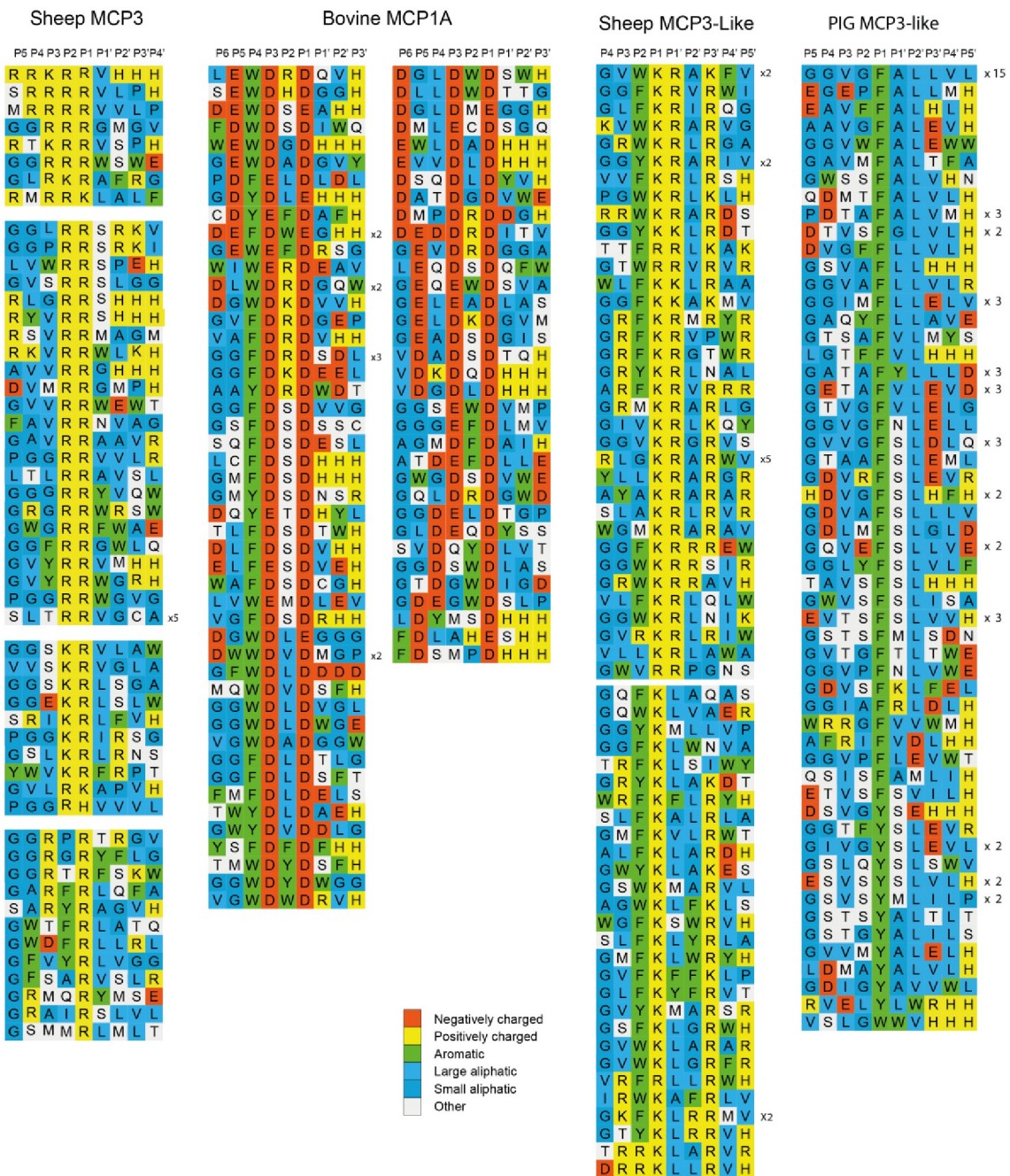

**Fig 6. Phage display analysis of sheep MCP3, cow MCP1A, sheep MCP3-like and pig MCP3-like.** After the last selection step of the phage display analysis, phages released by proteolytic cleavage of the three proteases were isolated and the sequences encoding the nonamers were determined. The general sequence of the T7 phage capsid proteins are PGG(X)₉HHHHHH, where (X)₉ indicates the randomized nonamers. The protein sequences were aligned into a P4-P3′ consensus, where cleavage occurs between positions P1 and P1′. If the sequence as found more than once this is indicated by the corresponding number to the left of the sequence. The amino acids are colour coded according to the side chain properties as indicated in the legend.

analysis of these sequences, a surprising pattern was observed (Fig 6). Many of the sequences contained multiple basic amino acids arranged in tandem arrays, primarily arginine. Several clones contained four arginine in a row (Fig 6). Clones with two or three arginine were the most common. However, clones with two arginine separated by one non-basic amino acid was also observed frequently. Almost no clones with only one basic amino acid were found and relatively few aromatic amino acids were observed within these sequences (Fig 6).

The next enzyme to be analysed by phage display was the bovine MCP-1A. From the chromogenic substrate assay, we had initial information about its P1 selectivity (Fig 5). Only the substrates with Asp in the P1 position and a negatively charged residue in the P3 position were cleaved, indicating that this enzyme prefers substrates with two negatively charged residues (Fig 5). Phage display of the bovine MCP-1A confirmed this result. Following seven rounds of selection, the sample with the enzyme had 5555-fold more phages than the control. In total of 120 plaques were picked for PCR amplification of the region, including the nine amino acid random region, and 96 of them, with the strongest and most clean PCR fragments, were sent for sequencing. As indicated from the chromogenic substrate assay, all the phage clones had at least two negatively charged residues and with a very high selectivity for Asp over Glu and, in general, the two negatively charged residues were separated by one amino acid. This separating amino acid showed very little selectivity except negatively charged amino acids were not favoured (Fig 6). A high preference for an aromatic amino acid in the P4 position was also observed (Fig 6). Several clones also had three negatively charged residues in the P1, P3 and P5 positions (Fig 6).

The third enzyme to be analysed by phage display was sheep MCP-3 like. For this enzyme, we had no information about its specificity as it did not cleave any of the chromogenic substrates. After several unsuccessful pannings, which indicated high selectivity, where the enzyme only reached 2–3 times over background we finally got a panning that after the 7th selection round gave a good selection and resulted in a 71-fold higher number of phages in the enzyme sample over background. A total of 120 plaques were picked and after PCR amplification of the region of interest 96 of the clones with strong and clean PCR fragments were sent for sequencing. After decoding the random region of the phages, a clear pattern emerged (Fig 6). The general pattern was two or three basic amino acids where the first was preceded by an aromatic amino acid and often with the sequence FKRLK or FKLAR (Fig 6). We also saw a striking absence of negatively charged amino acids in almost all the 70 sequences. In the few cases, where we saw a negative charged amino acid, they appeared upstream or downstream of this core sequence described above. The exact position of cleavage could not be determined by phage display and we had no information from the chromogenic substrate assay. This question was, however, later solved by the use of recombinant substrates (see below).

The fourth enzyme studied was the pig duodenase. From the chromogenic substrate assay this enzyme appeared to be a pure chymase with no tryptase or asp-ase activity. The phage display over six rounds of pannings resulted in a 111-fold higher number of phages in the enzyme sample compared to the PBS control. We picked 120 plaques and sent 96 of the most-clear PCR fragments for sequencing. The result showed that this enzyme was a classic chymase with a high selectivity for Phe or Tyr in the P1 position and also a selectivity for Leu in the P2´position (Fig 6). A high preference for aliphatic amino acids both upstream and downstream of the cleavage site was also observed (Fig 6). In the P1´position a preference for both aliphatic amino acids and Ser was observed (Fig 6). We could also see that the enzyme was relatively tolerant to negatively charged residues at many positions surrounding the cleavage sites, in marked contrast to the sheep MCP-3 like (Fig 6).

## Verifying the consensus sequence for sheep MCP3, cow MCP1A, sheep MCP3-like and pig MCP3-like using recombinant protein substrates

To verify the results from the phage display analysis, we used a new type of recombinant substrates that were developed in our lab. These substrates have been used in a number of previous studies and were shown to be highly reliable [29–35]. The consensus sequence, obtained from the phage display analysis, is in these substrates inserted in the linker region between two *E. coli* thioredoxin molecules by ligating a double-stranded oligonucleotide encoding the actual sequence into a BamHI and a SalI sites of the vector construct (Fig 7A). The thioredoxin molecule was used because of its tight folding, stability, relatively small size and high-level expression in *E. coli*. For purification, a His₆-tag was added to the C-terminal of these substrates (Fig 7A). Several related and unrelated substrate sequences were also produced with this system by ligating the corresponding oligonuclotides into the BamHI/SalI sites of the vector. All of these substrates were expressed as soluble proteins in *E.coli*, and purified on IMAC column to obtain 90–95% purity.

These recombinant proteins were used to study the preference of the sheep MCP3 for a panel of such substrates (Fig 7C–7E). The result shows that this enzyme very efficiently cleaves substrates with two, three or four positively charged amino acids in tandem (Fig 7C–7E). No or almost no cleavage was seen with substrates having only one basic amino acid or with no basic amino acids but containing aromatic amino acids. By using 10-fold more enzyme, a minor band appeared from one of the chymase substrates. This chymase substrate lacked positive charge and has instead a Phe in the P1 position (Fig 7G). The most optimal substrates were the ones with three or four Arg in tandem (Fig 7C). The substrates with four Arg in tandem show high degree of cleavage in the bacteria prior to purification and are thereby difficult to study due to high level of cleavage products in the 0 minutes time point. The substrate with only two Arg in tandem is 2–3 times less efficiently cleaved compared to the optimal triple Arg target (Fig 7C). Several additional variants were constructed to study the influence of amino acids following the three Arg residues of the most optimal substrate in panel C of the Fig 7. The substrate having LAA after the RRR was found to be 2–3 times more efficiently cleaved than the one with three Ala (AAA) (Fig 7D). In contrast, the ones with PAA and GAA were instead 10–15 times and 3–5 times less efficiently cleaved compared to the one with AAA, respectively (Fig 7D). A negatively charged residue directly N terminal of the three Arg residues (V**D**RRRAAAG) was also negatively affecting cleavage (Fig 7E). We could observe approximately 10-fold drop as compared to the variant 3 with the sequence VVRRRAAAG. Substrates with two Arg separated by one residue was found to be quite sensitive to the separating amino acid (Fig 7C and 7F). Phe residue between the two Arg residues was the preferred amino acid, however, with a close to 10-fold lower cleavage rate compared to the optimal three Arg variant, whereas the one with a Val or a Pro was almost not cleaved at all (Fig 7A and 7F). To determine in detail the difference in tryptase to chymase activity of the enzyme, we did a titration of the amount of enzyme needed to get the same amount of cleavage. By this analysis, we saw that the tryptase activity was ~160-fold stronger than the chymase activity based on the amount of enzyme needed to get roughly the same extent of cleavage (Fig 7H).

The next result from the phage display analysis to be verified was the result for the bovine MCP1A, the asp-ase. We here used a panel of substrates containing one or several negatively charged residues. The most optimal in this analysis was a substrate with three negatively charged residues, in positions P1, P3 and P5 (**EFDSD**GGLV), which also had been indicated from the phage display results (Figs 6 and 8A). The amino acids between the negatively charged residues in positions P2 and P4 were also of importance. An aromatic amino acid in P4 position seems to be highly favoured (Figs 6 and 8A). The result from the chromogenic

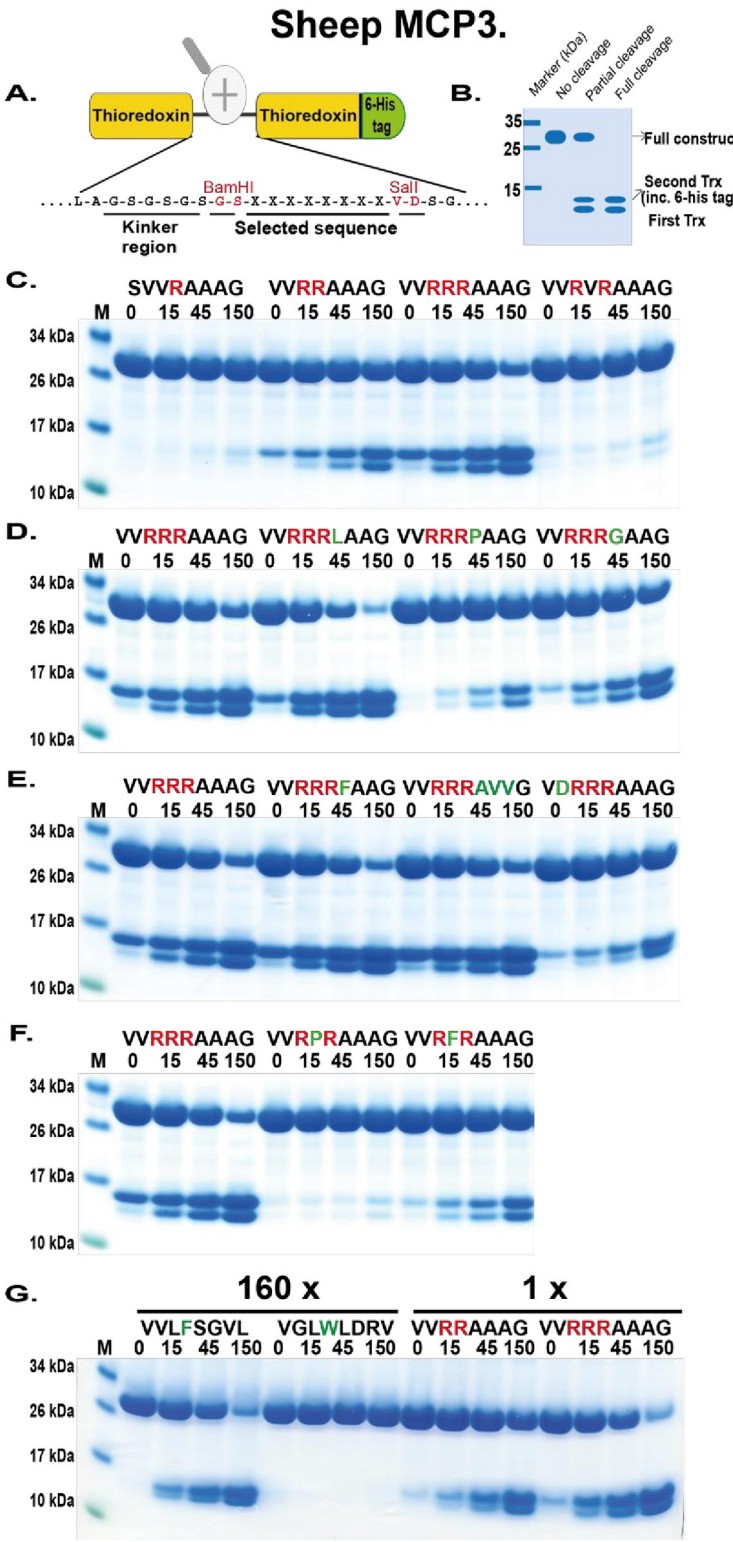

**Fig 7. Verification of the cleavage specificity of sheep MCP3 using recombinant protein substrates.** Panel A shows the overall structure of the recombinant protein substrates used for analysis of the efficiency in cleavage by the enzyme. In these substrates two thioredoxin molecules are positioned in tandem and the proteins have a His₆-tag positioned in their C termini. The different cleavable sequences are inserted in the linker region between the two thioredoxin molecules using two unique restriction sites, one Bam HI and one SalI site, which are indicated in the bottom of panel

A. In panel B, a schematic representation of a cleavage reaction is presented. In Panels C to G, the cleavage of several substrates by sheep MCP3 is presented. The sequences of the different substrates are indicated above the pictures of the gels. The time of cleavage in minutes is also indicated above the corresponding lanes of the different gels. The uncleaved substrates have a molecular weight of ~25 kDa and the cleaved substrates appear as two closely located bands with a size of 12–13 kDa. Residues of particular interest and that may differ between different substrates are marked in red or green for an easy identification.

substrate assay shows that negatively charged residues in positions P1 and P3 are essential for cleavage (Fig 5). Interestingly, there is also a relatively high selectivity for Asp over Glu in the P1 position (Fig 8C). Asp in the P1 position is cleaved at a rate 15–20 times more efficiently as compared to Glu in this position (Fig 8C). In the P2 position, Ser seems to be the most optimal. However, substrates with Arg or Leu in this position are also efficiently cleaved, whereas a larger Trp is less favoured (Fig 8B). The spacing of the negatively charged residues seems also to be important (Fig 8C and 8D). Negatively charged residues in the P1 and P4 positions were not favourable for cleavage. Rather, they needed to be in positions P1 and P3 for cleavage with an additional preference for negatively charged residue also in the P5 position.

The third phage display result to be verified was the one for sheep MCP3L. This is a highly selective tryptase with preference for several positively charged amino acids (Fig 6). From the phage display, Lys is highly favoured in what appears as the P1 position (Fig 6). Aromatic amino acids preceding this residue seem also be highly favoured as well as a positively charged residue in the P3´position. This result was verified with the 2xTrx substrates (Fig 9). A substrate having a Gly in the P3´position is cleaved although at a lower rate showing that the Arg in the P3´is not a sole residue for cleavage. This result also verifies the P1 position for this protease to be the first Lys residue (Fig 9E). Replacing Lys for Arg in the P1 position results in markedly lower cleavage rate showing the strong preference for Lys in this position (Fig 9B). A high selectivity is also seen in the P1´position. Arg, Leu and Lys are almost equally favoured whereas the small amino acid Gly almost completely prevents cleavage (Fig 9A). Phe in the P2 position is the most optimal residue but Tyr is also almost as efficiently cleaved, and based on phage display data, Trp also seems to be well accepted in this position (Figs 6 and 9C). In contrast, Gly in the P2 position seems to be unfavourable (Fig 9C). However, Gly in position P2 is favourable for cleavage if there is Arg in position P1´ and Leu in the P2´position (Fig 9C). Negatively charged residues, regardless of position, were not favourable for cleavage. No cleavage was seen when having an Asp in the P2´or P3´positions and very few negatively charged residues were present in the phage display sequences (Figs 6, 9D and 9E). Highly restricted selectivity of this protease was probably the main reason why no cleavage was detected with the panel of chromogenic substrates (Fig 5).

In contrast to the three enzymes previously described, the pig duodenase was a pure chymase with no tryptase activity (Fig 10A). It showed almost equal activity against Phe and Tyr-containing substrates but had low or no activity against Leu and Trp-containing substrates (Fig 10A, 10C and 10D). The pig duodenase seemed to have little preference for residues surrounding the Phe or Tyr in the P1 position except for what was observed from the phage display that aliphatic amino acids were preferred in most positions and that Leu was preferred in the P2´position (Fig 10C and 10D).

## Extended specificity of the two remaining cow duodenases (DDN1 and DDN1-like) by the use of recombinant protein substrates

To characterize the activities of the two remaining cow duodenases we decided to test them in a panel of recombinant substrates as described in the previous section. They both showed good activity on chromogenic substrates, making it easier to determine their extended

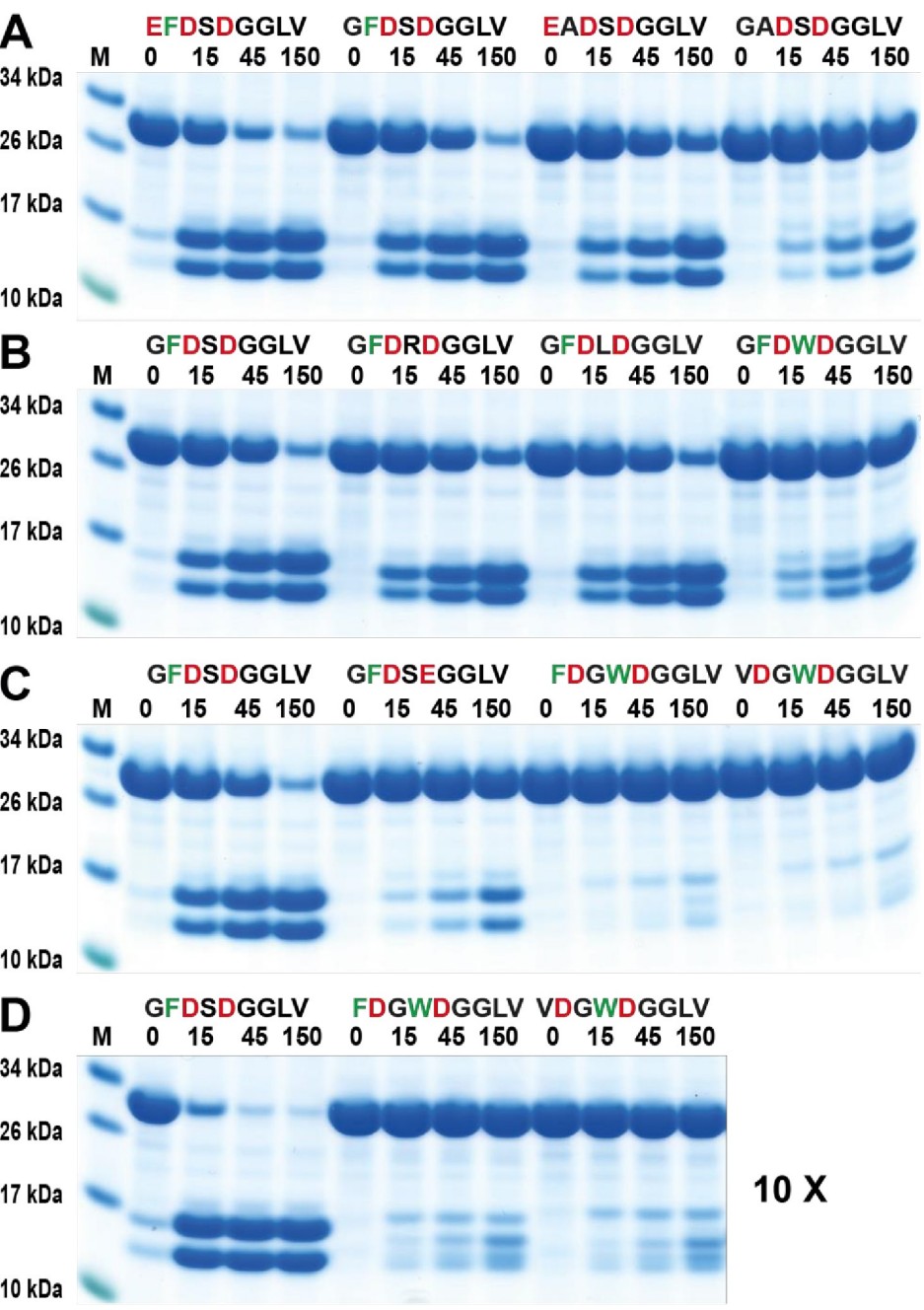

**Fig 8. Verification of the cleavage specificity of bovine MCP1A by the use of recombinant protein substrates.**
Panels from A to D show the cleavage of a number of substrates by bovine MCP1A. The sequences of the different substrates are indicated above the pictures of the gels. The time of cleavage in minutes is also indicated above the corresponding lanes of the different gels. The uncleaved substrates have a molecular weight of ~25 kDa and the cleaved substrates appear as two closely located bands with a size of 12–13 kDa. Residues of particular interest and that may differ between different substrates are marked in red or green for an easy identification. In panel D, we used 10-fold more enzyme compared to the other cleavage reactions.

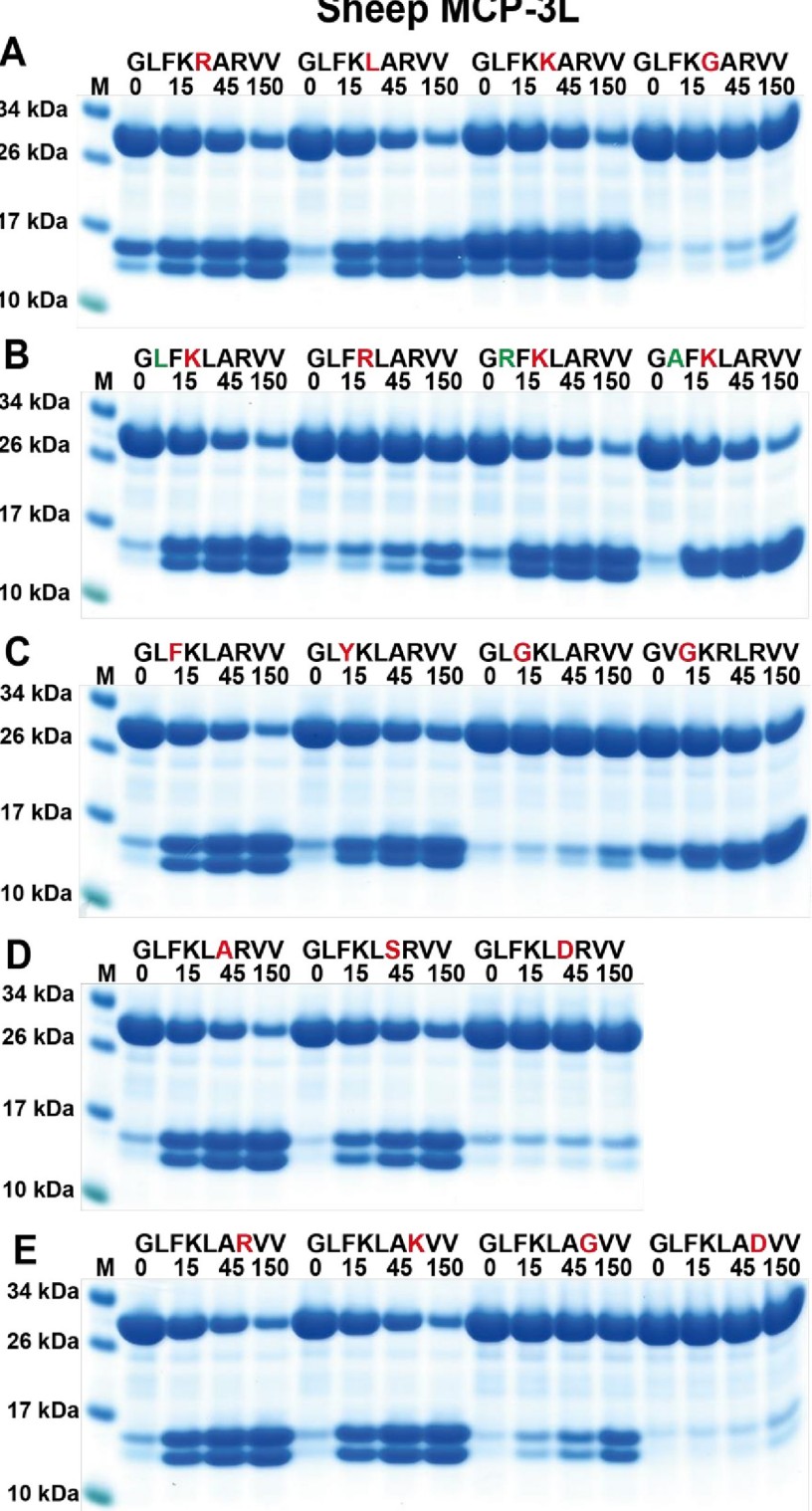

**Fig 9. Verification of the cleavage specificity of sheep MCP3-like using recombinant protein substrates.** Panels A to E show the cleavage of several substrates by MCP3-like. The sequences of the different substrates are indicated above the pictures of the gels. The time of cleavage in minutes is also indicated above the corresponding lanes of the different gels. The uncleaved substrates have a molecular weight of ~25 kDa and the cleaved substrates appear as two closely located bands with a size of 12–13 kDa. Residues of particular interest and that may differ between different substrates are marked in red or green for an easy identification.

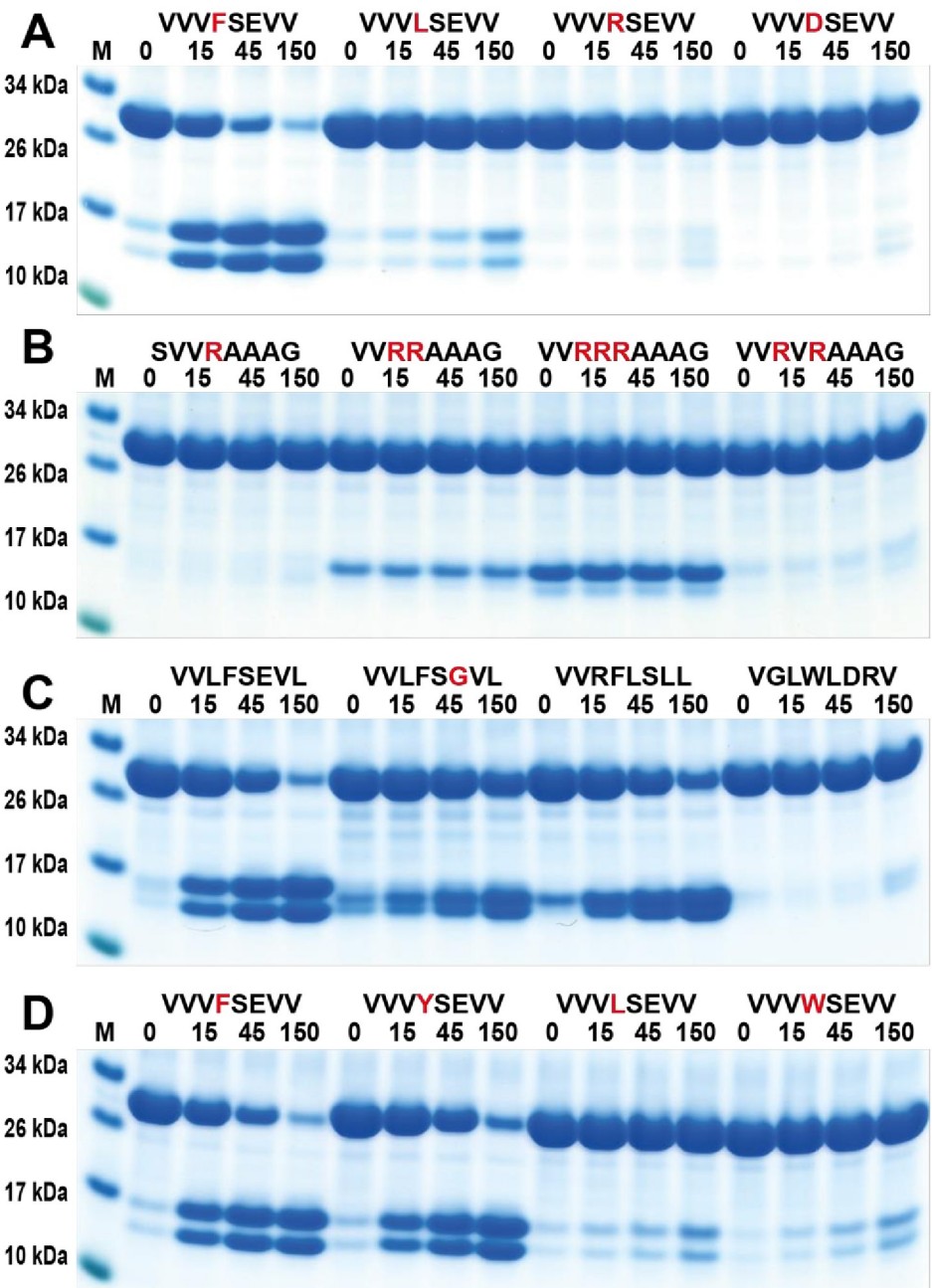

**Fig 10. Verification of the cleavage specificity of pig MCP3-like using recombinant protein substrates.** Panels A to D shows the cleavage of several substrates by the pig MCP3-like. The sequence of the different substrates are indicated above the pictures of the gels. The time of cleavage in minutes is also indicated above the corresponding lanes of the different gels. The uncleaved substrates have a molecular weight of ~25 kDa and the cleaved substrates appear as two closely located bands with a size of 12–13 kDa. Residues of particular interest and that may differ between different substrates are marked in red for an easy identification.

specificities with a panel of recombinant substrates. Both the bovine DDN1 and DDN1-like enzymes seemed from the chromogenic substrate assay to be true dual enzymes with both tryptic and chymotryptic activities (Fig 5). We first tested a set of four substrates with different amino acids in the P1 position, Phe, Leu, Arg and Asp as marked in red in the Fig 11. Bovine DDN1 showed good activity on both chymase, leu-ase and tryptase substrates but no activity toward a substrate with an Asp residue in the P1 position. The results indicate that the enzyme has dual specificity being active toward both chymase and tryptase substrates (Fig 11A). Interestingly, with a different set of amino acids N and C-terminally of the Arg residue, a single Arg also showed low rate of cleavage. However, substrates with two or three Arg residues in tandem were more efficiently cleaved (Fig 11B). Very low activity was also seen with two Arg residues separated by a Val (Fig 11B). The chymase activity was the highest with Phe-containing substrates but also moderate with Tyr and Leu-containing substrates. However, tryptophan was found not to be efficient in P1 position (Fig 11C and 11D). The most optimal chymase substrate contained the sequence VVRFLSLL and the best tryptase substrate had three Arg residues, VVRRRAAAG (Fig 11B and 11C, respectively).

Bovine DDN-like protease was found to have a very similar specificity as bovine DDN1, however with some differences (Fig 12). With the first four substrates, the bovine DDN-like showed the best activity against the tryptase substrate and relatively low activity against both the Phe and the Leu-containing substrates (Fig 12A). The activity on the second set of substrates showed activity on the single Arg containing substrate but also here better activity against the substrates with two and three Arg residues, and also some activity against the substrate with two Arg residues separated by a Val (Fig 12B). The chymase activity was highly dependent on surrounding amino acids where Leu and Arg were favoured over Val in the P2 and P1´positions (Fig 12A and 12C). Like bovine DDN1, a tryptophan in the P1 position was not a preferred substrate (Fig 12C and 12D). As to its chymase activity, bovine DDN-like, similarly to bovine DDN1, prefers a Phe in the P1 position over Tyr and Leu (Fig 12D).

## Bioinformatic screening of cow and sheep proteomes for potential *in vivo* targets of these enzymes

To try to identify potential *in vivo* targets for these enzymes, we screened the cow proteome with the consensus sequence for the cow asp-ase. The asp-ase is one of the more specific of the six analysed proteases, which increase the chance to identify a potential target. We obtained several very interesting targets. The one that we think is the most interesting was the identification of five almost consensus sites in cow mucin-5B, one of the major salivary mucins, in the following positions; 833–840 EWFDVDYP,1293–1300 EWFDVDYP, 1897–1905 EWFDVDYP, 2446–2453 EWFDVDYP, 3057–3064 EWFDVDYP, 3590–3597 EWFDVDFP (Fig 13). Cows produce between 50 and 120 litres of saliva per day and mucin-5B is there to facilitate digestion by lubricating the ingested material. It is possible that quite a lot of these mucins escape digestion during passage of the abomasum, the true stomach, where most of the protein is being digested by pepsin, due to the very heavy glycosylation of the protein core. The cow may thereby have a need for additional enzymes to ensure efficient recycling of the amino acids of the saliva. We also identified several cell adhesion molecules among the potential targets of the cow asp-ase, indicating that it also takes part in regulating intestinal permeability. However, these cell adhesion molecules primarily belong to the protocadherin FAT family including FAT1, FAT2, FAT3 and FAT4 but also cadherin 16 and 23 with multiple potential cleavage sites. Collagen alpha-1 (VII and XXII isoforms), collagen alpha-3 (VI), laminin, dysferlin, fibrillin-3, desmocollin-1 and desmoplakin were also identified as potential targets. Many of these are more broadly expressed and we do not see them as the most likely

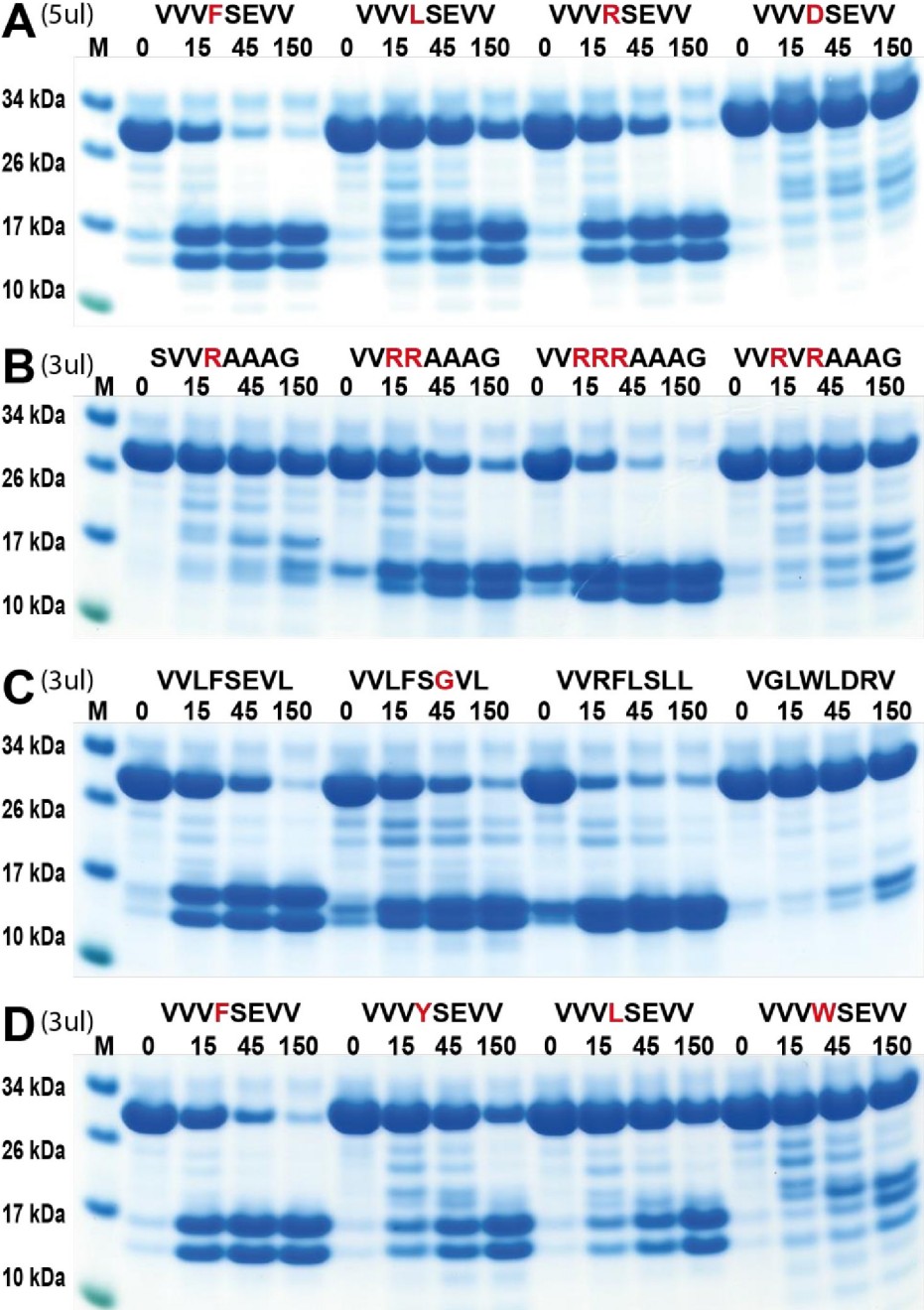

**Fig 11. Analysis of the cleavage specificity of bovine duodenase DDN1 using recombinant protein substrates.**
Panels A to D shows the cleavage of several substrates by the bovine DDN1. The sequence of the different substrates are indicated above the pictures of the gels. The time of cleavage in minutes is also indicated above the corresponding lanes of the different gels. The uncleaved substrates have a molecular weight of ~25 kDa and the cleaved substrates appear as two closely located bands with a size of 12–13 kDa. Residues of particular interest and that may differ between different substrates are marked in red or green for an easy identification.

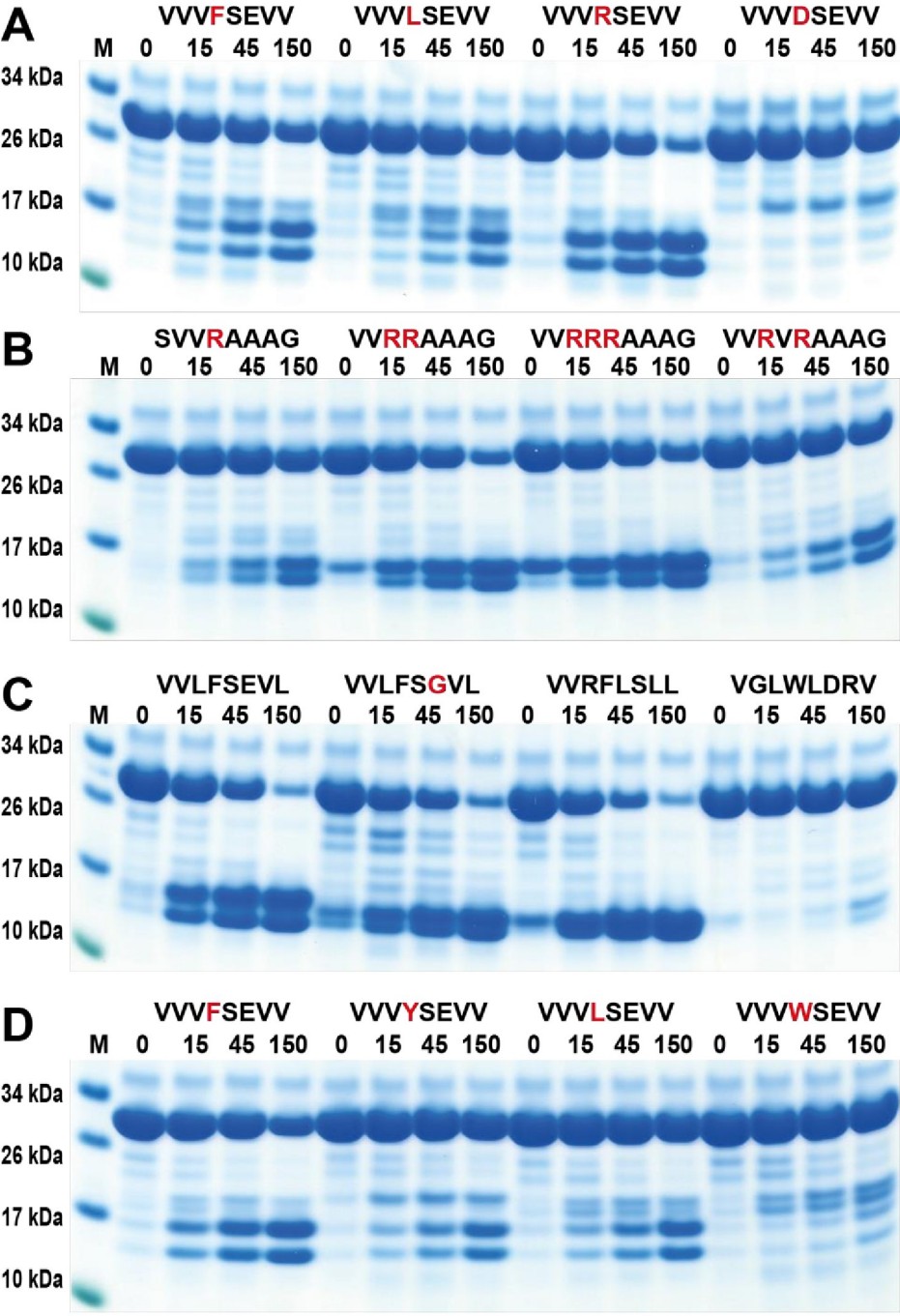

**Fig 12. Analysis of the cleavage specificity of bovine duodenase DDN1-like using recombinant protein substrates.** Panels A to D shows the cleavage of several substrates by the bovine DDN-like. The sequence of the different substrates are indicated above the pictures of the gels. The time of cleavage in minutes is also indicated above the corresponding lanes of the different gels. The uncleaved substrates have a molecular weight of ~25 kDa and the cleaved substrates appear as two closely located bands with a size of 12–13 kDa. Residues of particular interest and that may differ between different substrates are marked in red for an easy identification.

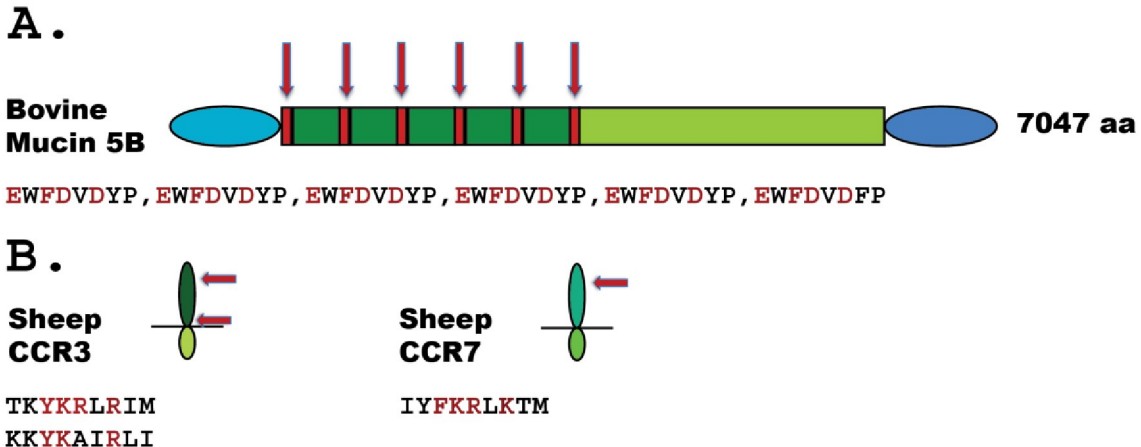

**Fig 13. Schematic representation of three potential targets for bovine MCP1A and sheep MCP3-like.** In panel A, we show a schematic picture of bovine Mucin 5B with the different regions of the protein marked in different colours and where the six consensus sites for bovine MCP1A are marked by red arrows. The sequence of the six potential cleavage sites are listed below the schematic figure. In panel B, we show the schematic structure of two chemokine receptors having potential cleavage site for sheep MCP3-like. The position of the membrane is marked with a black line and the potential cleavage sites by red arrows. The sequences of the potential cleavage sites are listed below the schematic figure.

biologically relevant targets for these enzymes. A detailed list of potential cleavage sites in these molecules are found in a S1 Table.

The next enzyme to study for its potential targets was one of the sheep enzymes that showed a high specificity, sheep MCP3L. This enzyme prefers targets having the sequence F/YKXXR/K and no negatively charged residues close to the cleavage site. After screening the sheep proteome, we found several near consensus sites in several receptors to inflammatory cytokines and chemokines, among them the receptors for the chemokine receptors 3 and 7 (Fig 13). None of these were found in the list for the bovine enzyme. The fact that we found two receptors for important inflammatory chemokines suggests that this duodenase limits excessive inflammation caused by the massive amounts of bacterial products entering the small intestine in these animals.

## Discussion

Here we present the primary and extended specificity of six structurally closely related duodenases, two from sheep, three from cows and one from pigs. Interestingly, their catalytic specificity was strikingly different, two are potent tryptases, one an asp-ase, two have potent dual tryptase and chymase activity and one is a potent chymase with no tryptase activity. These six duodenases have most likely appeared after successive gene duplications from a granzyme or a cathepsin G gene (Fig 2). So far, duodenases have only been found in the genomes of ruminants and pigs (Fig 2) [13]. Interestingly, immunochemistry data show that the cow duodenases are expressed primarily by non-hematopoietic cells in the intestinal mucosa, the Brunner's glands [15]. These enzymes appear also to be secreted into the duodenal lumen and thereby thought to be involved in food digestion and not primarily in immune defence as the majority of other chymases [36]. These proteases have also been shown to represent the absolute majority of endopeptidase activity of the duodenal mucosa and constitute as much as 0.3–0.4% of the total protein content of a duodenal homogenate [14]. When we compared the primary sequences of these duodenases we saw that they were highly homologous (Fig 14). Interestingly, only minor changes in the primary sequence can result in relatively large differences

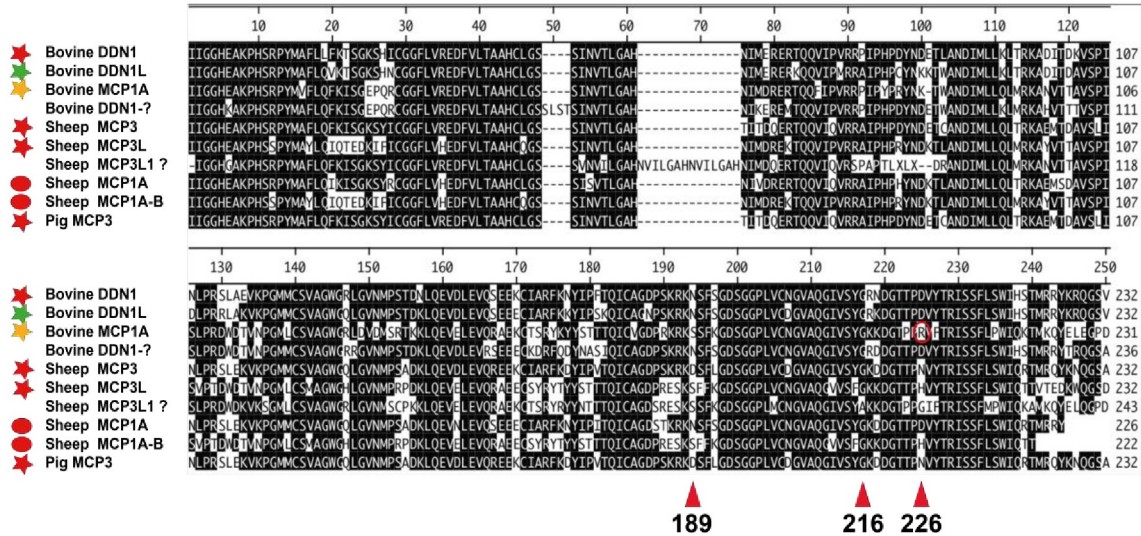

**Fig 14. Sequence alignment of a panel of cow, sheep and pig duodenases.** The figure shows the sequence alignment where three residues of importance for target selectivity are highlighted, residues 189, 216 and 226 (chymotrypsin numbering). These three residues form the catalytic pocket and as can be seen cow MCP1A differ from the other duodenases in position 226 where it has an Arg, which is marked by a red circle. This position is of major importance for its different primary specificity as being an asp-ase. Sheep DDN1-like lacks the N terminal part of the mature protease and is most likely a pseudogene.

in both primary and extended cleavage specificities of these enzymes. As can be seen from Fig 14, a change in position 226 in the bovine MCP-1A (marked by a red circle) is most likely the primary reason for the change in specificity of this enzyme to a strict asp-ase (Fig 14). This shows the large plasticity in this family of proteases to relatively rapidly adapt to changes in feeding habits and other changes in life style.

Ruminants have a highly specialized digestive system adopted for the digestion of cellulose rich food sources as grass and other plants, and cellulose is very difficult or impossible to digest by mammals due to the lack of endogenous cellulases. This adaptation for cellulose rich food has most likely involved several adaptive changes to their digestive system. One important factor here is that mammals do not express cellulases and thereby need to obtain help from bacteria, protozoa and fungi to take advantage of this energy rich food source. In order to do so, another remarkable set of gene duplications have occurred in another locus that also primarily was adapted for immune functions, the lysozyme C gene locus (Fig 1). In humans we have only one gene for a classical lysozyme, in mice and rabbit there are two of which one is expressed primarily in immune cells and the second gene primarily in the intestinal region taking part in food digestion [1,37]. Interestingly the rat also has two lysozyme c genes, however, one appears to have been silenced and the remaining copy is expressed in both the small intestine and in macrophages [38]. Cows and sheep have in this locus experienced a series of gene duplications. Cows have now ten functional lysozyme genes of which at least four are expressed primarily in the intestinal region and there taking part in food digestion (Fig 1) [2,3]. Sheep has undergone a similar expansion and have seven lysozyme genes, whereas pigs have only one, similar to humans (Fig 1). This massive expansion in cow and sheep must have occurred after the separation of sheep and cows from the common ruminant ancestor of pigs, cows and sheep. Interestingly, also is the expression pattern of the intestinal lysozymes. The cow intestinal lysozyme genes are expressed primarily in the true stomach, the mouse and rat lysozymes in the small intestine and the rabbit in the colon [1,37,39]. The single pig lysozyme is expressed both in stomach and non-stomach tissues [40]. Langur, a leaf eating primate, also

express high levels of lysozyme in the stomach and they also have only one lysozyme c genes as other primates [41]. This shows the large flexibility in solving a problem by different organisms. The problem can be solved by changing tissue specificity and most likely increase expression, as seen in rats and langur, or by increase gene number and tissue specificity of some of the newly generated copies as seen in ruminants, mice and rabbits. Interestingly, a similar situation is observed in a leaf eating bird, the hoatzin, which also have experienced a massive expansion of the lysozyme locus to obtain a number of additional copies of genes now expressed in their stomach [42]. At least six very closely related genes encode stomach expressed genes and the hoatzin also have three non-stomach traditional lysozyme genes whereas closely related birds not having a leaf diet like pigeons and cuckoos have only one gene homologous to these lysozyme genes in the hoatzin [42]. These findings give strong indications for several independent gene duplication events in different animal species resulting in similar adaptations to cellulose rich food.

The cow stomach lysozyme genes seem primarily to be secreted in the anterior part of the true stomach, the last of the four bovine stomachs, and reach very high concentrations (100 ug/ml) in the fluid content of the true stomach [39]. In this part of the true stomach the pH is around 6 and then drops towards the posterior part to more optimal pH 2 for pepsin to then again return to pH 5–6 when entering the duodenum [39]. The cow lysozymes have developed a high resistance to pepsin and have a pH optimum around 5 why they may be active both in the anterior part of the true stomach and in the duodenum to strip bacteria from their cell wall and make them accessible as food source [39]. Interestingly, the rabbit intestinal lysozyme gene is expressed only in the distal part of the colon [37]. The rabbit produces two types of faeces one soft containing high levels of mucus and with very high levels of lysozyme and one more solid hard faeces that is low in lysozyme and mucus. The soft faeces are re-ingested and processed through the stomach to absorb nutrients from the bacterial cells that process fibre rich food in the cecum and proximal colon [37]. The final result in the rabbit is thereby very similar of the bacterial symbiont-aided digestion of fibre to what is seen in cows but performed by completely different mechanisms, by foregut and hindgut fermenting respectively [37]. The rabbit appears, together with hares and picas to be relatively unique in this particular type of hindgut fermentation, and this has been named cecotrophy [37].

Lysozyme and their genes are a very interesting also from a molecular evolutionary viewpoint as lysozyme is a member of a larger family of related genes with very different functions. At least eight different lysozyme related genes have been identified and also three different types of peptidoglycan cleaving lysozymes, lysozyme c, g and i [43,44]. Lysozyme c is the classical lysozyme of mammals involved in immunity by its potent enzymatic activity against the bacterial cell wall [39]. This gene is most likely one of the most ancient gene of all the lysozyme related genes in humans and is found in diverse vertebrate and non-vertebrate species including insects and at least one crustacean, a prawn [44]. Another well-known member of the lysozyme related gene family is the lactalbumin found in the milk of essentially all mammals where it is important for the formation of lactose [43]. Lactalbumin has a nearly identical three-dimensional structure as lysozyme although it only has 40% amino acid sequence identity [43]. It lacks bactericidal activity and is expressed in the lactating mammary glands, where it binds a calcium ion and modifies the activity of β-galactosy-transferase-1, so that the complex catalyzes the synthesis of lactose. Lactalbumin with its present mammalian function in the formation of lactose is only found in mammals where it is widely spread. This duplication made it possible for the early mammals to form a nutrient rich food for their offspring, a mechanism that successfully has been maintained in essentially all mammals of today.

The new lysozyme c genes of ruminants seem to have appeared during early ruminant evolution, which has been estimated to have occurred around 55 million years ago [40]. This has

been based on the finding that pigs only have one lysozyme c gene and all ruminants studied having essentially high numbers of lysozyme c type genes, including cow, water buffalo, yak, Tibetan antelope, zebu, goat and sheep (Fig 1) [2]. Interestingly gene duplications following the appearance of the eight different lysozyme related genes seem primarily involve the lysozyme c genes in ruminants and a duplication in rodents. When we look at the several of the other members of this large family only rarely a single duplication has occurred as the case for the LysL1 and genes in primates and Lys like 6 in dogs (S1 Fig). The only massive expansion seems to be this increase from one to seven and ten genes in the lysozyme c genes in sheep and cows (Fig 1).

In addition to being expanded in number and having changed tissue specificity the new lysozyme c genes of ruminants have accumulated a number of additional important structural changes. Such structural modifications are their resistance to denaturation by the low pH of some part of the intestinal region and the increase in resistance to cleavage by pepsin [2].

The functional significance of the new lysozyme C copies is easy to understand as this enables the cows to more efficiently use cellulose rich food, by first allowing the bacteria to break down the cellulose and to build new macromolecules which then can become an energy rich food for the cows. The cows use the lysozyme to kill and open the bacteria and make use of their proteins, fatty acids and sugars. However, the functions of the duodenases, which have appeared and expanded what it seems in parallel with the lysozyme genes are less clear. The two dual specificity cow duodenases, DDN1 and DDN-like, and the pig chymase are most likely similar to trypsin and chymotrypsin classical digestive enzymes. However, the role of the more specific proteases, sheep MCP-3, bovine MCP-1A and sheep MCP-3-like are more difficult to envisage at present. Are they like enterokinase involved in activation of other enzymes or completely new functions and why are there such a big difference in the specificity of these enzymes between cow and sheep? Upon initiating this study, we expected to find a major similarity in primary and extended specificities between the enzymes of the three different species. However, to our surprise this was not the result, instead we here see a broad panel of different primary and extended specificities and also enzymes with high selectivity and such with a broad and relatively unselective target specificity. A broad bioinformatics screening of the cow and sheep proteome for potential targets for the two more selective enzymes resulted in potentially very interesting substrate candidates. One of them is mucin-5B, one of the major mucins of the saliva. Cows produce massive amounts of saliva to facilitate ingestion and lubricate the ingested material. It has been estimated that they produce between 50 and 120 litres of saliva per day and it is possible that a relatively large fraction of these highly glycosylated proteins escape digestion during passage through the true stomach, the abomasum [45]. It is possible that the duodenases cleave and thereby facilitate reabsorbtion of the energy stored in the remaining mucin 5B entering the duodenum.

One of the sheep tryptase-type duodenases also showed a high target selectivity and the screening for potential substrates in the sheep proteome resulted in the identification of two chemokine receptors as potential prime targets. We have so far no direct evidence for sheep enzyme cleaving these two potential targets, but the fact that two such targets were identified among the top hits make it intriguing to consider this function for sheep duodenase. The massive amounts of bacterial products entering the duodenum after lysozyme mediated digestion of intestinal bacteria in the foregut fermenting ruminants compared to other mammals may result in a strong TLR receptor activation and thereby an excessive inflammatory response that needs to be balanced. The sheep MCP3 may here have this as major function.

Although both immunohistochemistry and immunogold staining show expression almost exclusively of the cow duodenases to the Brunner´s glands of the duodenum and not to hematopoietic cells in this tissue, other studies also indicate expression of these duodenases to

mast cell [15,17,19,20]. These proteases may therefore have functions both in food digestion and in immunity, a question that needs to be addressed in more detail to understand their apparent complex role in ruminant biology.

In summary, a remarkable diversity in both primary and extended specificities have been found among these six duodenases. For the enzymes with a relatively broad specificity, like the two bovine dual specific enzymes and the pig broad spectrum chymase, it is not easy to see the advantage of having them in food digestion as the pancreas supplies the duodenum with a multitude of broad-spectrum proteases including trypsin, chymotrypsin and pancreatic elastases. Thus, we may wonder why additional proteases are needed. In marked contrast, for the more specific duodenases including the cow asp-ase and the highly specific sheep tryptase, we found potential targets that could physiologically make a lot of sense. Mucin-5B is produced in large amounts to lubricate the food during rumination and its digestion is difficult by the traditional digestive proteases. Thus, additional specific enzymes such as chymases may be needed to recover the amino acids of the saliva. The large amount of bacterial break-down products entering the small intestine of foregut fermenters may also be a problem due to their strong inflammatory properties. New enzymes may therefore be needed to balance excessive TLR, Nod and RIG receptor activation. An enzyme cleaving chemokine receptors appears a suitable candidate, so both mucin-5B and these chemokine receptors are likely to be some of the targets for these enzymes. However, this is no doubt not the full scope of activities for these chymases. What are the additional targets and do the broad-spectrum duodenases also have specific regulatory functions, and do they also have additional functions in mast cells? These are questions that need to be addressed to give us a more complete picture of the biology of this fascinating new subfamily of digestive and potentially immunomodulatory enzymes.

# Materials and methods

## PCR analysis of cow intestinal region for the expression of duodenases

Intestinal tissue sample from cow duodenum was obtained from a slaughter house in Uppsala. Total RNA was prepared by the guanidine-isothiocyanate method and single cDNA was prepared to be used as template for the PCR analysis [46]. A set of primers for the five individual duodenase genes identified in the cow genome deposited in Genbank, at the time, was designed and ordered from Sigma-Aldrich (Darmstadt, Germany) PCR was run under standard conditions, Start 95˚C for 2 minutes, then 45 cycles with 95˚C for 30 seconds followed by 60˚C for 30 seconds and then by 72˚C for 1 minute. The final run was followed by a chase at 72˚C for 10 minutes. The samples were then separated on 2% agarose gels in TEB buffer.

The sequence of the PCR primers used to study the presence of duodenases in bovine duodenum are as follows- Forward primers also encode His-6 tag and enterokinase site:

BDMD1 forward: CATGAGCGGCCGC**CATCACCATCACCATCAC**GACGATGACGATAAGA TCATCGGGGGGTCACGAGGCCAAG

BDMD1 reverse: CAGCTCGAGTCCACCCTGAGCACACATCACA

BDMD2 forward: CATGAGCGGCCGC**CATCACCATCACCATCAC**GACGATGACGATAAGA TCATTGGGGGGCATGATGCAAG

BDMD2 reverse: CAGCTCGAGACCTCAGGCTGATCCCTACCA

BDMD3 forward: CATGAGCGGCCGC**CATCACCATCACCATCAC**GACGATGACGATAAGA TCATCGGGGGCCACGAGGCCAAG

BDMD3 reverse: CAGCTCGAGGTCCACCCCAAGGACACGT

BDMD4 forward: CATGAGCGGCCGC**CATCACCATCACCATCAC**GACGATGACGATAAGA TCATCGGGGGCCACAAGGCCAAG

BDMD4 reverse: CAGCTCGAGAGTCCATCCCGAGGACATGTCAT

BDMD5 forward: CATGAGCGGCCGC**CATCACCATCACCATCAC**GACGATGACGATAAGA
TCATCGGGGGCCACGAGGCCAAG
BDMD5 reverse: CAGCTCGAGGTCCTTGGAGAAACCTCAGTCTGG

## Phylogenetic analyses

For all proteases, the entire sequence of the active form, not including the signal sequence and activation peptide were used in the multiple alignments. The phylogenetic analyses were performed using a Bayesian approach as implemented in MrBayes version 3.2.7a. Markov Chain Monte Carlo (MCMC) analyses were used to approximate the posterior probabilities of the trees. Analyses were run using the MrBayes manual standard protocol [47]. The phylogenetic trees were drawn in FigTree 1.4.2 (http://tree.bio.ed.ac.uk/software/figtree/). This is a smaller analysis using the same strategy as in a previous article on the evolution of the hematopoietic serine proteases [13]. All the accession numbers for the sequences used in this study is listed here below. Bovine-MCP-2 (XP_002696732), Bovine MCP-2 (XP_002696686), Bovine DDN1 (NP_776721), Bovine MCP1A (XP_002696733), Bovine CTSG (XP_002696688), Bovine CTSG (XP_015314760), Bovine GzmH (XP_002696734), Bovine DDN1_like (XP_003587611), Bovine DDN (XP_002696691), Bovine GzmH (XP_010815453), Bovine GzmB (XP_002696692), Sheep MCP2 (XP_027813303), Sheep MCP3-Like (XP_027813059), Sheep GzmH-Like (XP_027813010), Sheep MCP2-like (XP_027813014), Sheep MCP1A (NP_001009472), Sheep MCP2_like (NP_001116477), Sheep MCP3 (NP_001009411), Sheep CTSG-Like (XP_014957329), Sheep CTSG-Like (XP_027813305), Sheep GzmH-Like (XP_027813015), Sheep MCP1A (XP_027813306), Sheep MCP3-like (XP_027813016), Sheep GzmB-Like (XP_027813307), Sheep GzmB-Like (XP_027813308), Pig MCP3 (XP_003482301), Pig CTSG (XP_001926799), Pig CTSG-like (XP_013833608), Pig GzmH (NP_001137165), Pig GZmB (NP_001137182), Human Cma1 (NP_001827), Human CTSG (NP_001902), Human GzmH (NP_219491), Human GzmB (NP_004122), rMCP-5 (NP_037224), Rat CTSG (NP_001099511), Rat GzmB (NP_612526), mMCP-5 (NP_034910), Mouse CTSG (NP_031826), Mouse GzmB (NP_038570), Human Coagulation Factor X (NP_001299603), Human Complement Factor B (NP_001701), Mouse Coagulation Factor X (NP_031998), Platypus Coagulation Factor X (NP_001121086).

## Production and purification of recombinant sheep, cow and pig duodenases

The sequences of the sheep, cow and pig duodenases were retrieved from the Genbank databases. The cDNA sequences for two sheep enzymes, the sheep duodenase (Chymase III now named MCP3) were subcloned into a pAcGP67B vector that encodes a secretion signal, ubiquitin and an enterokinase (EK) cleavage sequence immediately before cloning site as previously described [27]. The resulting expression vectors were transfected into baculovirus-infected insect cells (High Five™) (Invitrogen, Carlsbad, CA). Purification was performed on heparin–Sepharose (GE Healthcare, Piscataway, NJ). The enzymes were activated by EK cleavage (Roche, Nutley, NJ) followed the protocol already described in detail [27]. After activation, the second heparin–Sepharose column was used to remove EK and released N-terminal peptides.

The sequences for the cow duodenases were cloned from cDNA obtained from calf small intestines. Fresh calf small intestinal tissue samples were obtained from a slaughterhouse in Uppsala and kept on ice until RNA was prepared by the Guanidine-Isothiocynanate method [46]. PCR primers were designed based on the gene sequences retrieved from GenBank and used in a standard PCR reaction of 35 cycles. Three of the five primer pairs generated strong

PCR fragments corresponding to the genes DDN1, DDN1L and MCP1A. These PCR fragments were cleaved with the restriction enzyme EcoRI and XhoI and ligated into the mammalian expression vector pCEP-Pu2 [48]. Following transfection and purification on Ni chelating IMAC columns according to previously described protocols, the protein purity and concentration was estimated by separation on 4–12% pre-cast SDS-PAGE gels (Invitrogen, Carlsbad, CA, USA) [21,48]. Protein samples were mixed with sample buffer, and β-mercapto-ethanol was added to a final concentration of 5%. To visualize the protein bands, the gel was stained with colloidal Coomassie Brilliant Blue [49]. The sheep and pig duodenase sequences were retrieved from the Genbank databases and ordered as designer genes from GenScript (Piscataway, NJ, USA) who also cloned these constructs into the episomal vector pCEP-Pu2.

## Chromogenic substrate assay

To determine the primary specificity of the different duodenases, eleven chromogenic substrates were tested for their sensitivity to cleavage by the duodenases and by the human mast cell chymase and by thrombin as reference proteases. We used the following eleven substrates: Suc-AAPF-pNA, Suc-LLVY-pNA, Suc-AAPI-pNA, Suc-AAPA-pNA, Suc-AAPL-pNA, Suc-AAPV-pNA, Suc-VLGR-pNA, Suc-GPR-pNA, Suc-YVAD-pNA, Suc-VEID-pNA and Suc-IEPD-pNA from Bachem (Bubendorf, Switzerland) and Chromogenix (Mölndal, Sweden). Reactions were prepared in 96 well microtiter plates, to which 5 μl of substrate (0.2 mM final concentration), a few μl activated enzyme depending on activity, and PBS was added to a final volume of 200 μl. The reaction was done at 20˚C with measurements taken spectrophotometrically with a Versa-max microplate reader (Molecular Devices, Sunnyvale, CA) at 405 nm at 0, 20, 40, 60, 120, 180, 240, 300 and 360 minutes. Reactions were done in triplets together with a with a blank to which no enzyme was added. Results were then graphed by subtracting the blank measurement at each time point and using the mean of the three reactions.

## Determination of cleavage specificity by phage-displayed nonapeptide library

A library of $5x10^7$ unique phage-displayed nonameric peptides was used to determine the cleavage specificity of the cow and sheep duodenases as previously described [50–52]. In these T7 phages, the C-terminus of the capsid protein 10 were manipulated to contain a nine aa long random peptide followed by a $His_6$-tag [50]. An aliquot of the amplified phages (~$10^9$ pfu) were bound to 100 μl Ni-NTA beads by their $His_6$-tags for 1 h at 4˚C under gentle agitation. Unbound phages were removed by washing ten times in 1.5 ml 1 M NaCl, 0.1% Tween-20 in PBS, pH 7.2, and two subsequent washes with 1.5 ml PBS. The beads were finally resuspended in 1 ml PBS. Activated protease was added to the resuspended beads and left to digest susceptible phage nonapeptides under gentle agitation at room temperature overnight. PBS without protease was used as control. Phages with a random peptide that was susceptible to protease cleavage were released from the Ni-NTA matrix, and the supernatant containing these phages was recovered. To ensure that all released phages were recovered the beads were resuspended in 100 μl PBS (pH 7.2) and the supernatant, after mixing and centrifugation, was added to the first supernatant. To ensure that the $His_6$-tags had been hydrolyzed on all phages recovered after protease digestion, 15 μl fresh Ni-NTA agarose beads were added to the combined phage supernatant and the mixture agitated for 15 min followed by centrifugation. A control elution of the phages still bound to the beads, using 100 μl 100 mM imidazole showed that at least 1 x $10^8$ phages were attached to the matrix during each selection. Ten μl of the supernatant containing the released phages was used to determine the amount of phages detached in each round of selection. Dilutions of the supernatant were plated in 2.5 ml 0.6% top agarose

containing 300 μl of *E. coli* (BLT5615), 100 μl diluted supernatant and 100 μl 100mM IPTG. The remaining volume of the supernatant was added to a 10 ml culture of BLT5615 (OD ~0.6). The bacteria had 30 min prior to phage addition been induced to produce the T7 phage capsid protein by the addition of 100 μl 100 mM IPTG to the culture. The bacteria lysed approximately 75 minutes after phage addition. The lysate was centrifuged to remove cell debris and 500 μl of the phage sub-library was added to 100 μl fresh Ni-NTA beads, to start the next round of selection. After binding the sub-library for 1 h at 4˚C under gentle agitation, the Ni-NTA beads were washed 15 times in 1.5 ml 1 M NaCl, 0.1% Tween-20 in PBS, pH 7.2, followed by two subsequent washes with 1.5 ml PBS.

Following five to seven rounds of selection, 120 plaques were isolated from LB plates after plating in top agarose. Each phage plaque, corresponding to a phage clone, was dissolved in phage extraction buffer (100 mM NaCl and 6 mM $MgSO_4$ in 20 mM Tris-HCl pH 8.0) and vigorously shaken for 30 minutes in order to extract the phages from the agarose. The phage DNA was then amplified by PCR, using primers flanking the variable region of the gene encoding the modified T7 phage capsid-protein. The quality and quantity of the amplified DNA was determined by gel electrophoresis and the 96 samples with best DNA quality were sent in a microtiter plate for sequencing to GATC Biotech in Germany (now Eurofins, Ebersberg, Germany).

## Generation of recombinant substrates for the analysis of the cleavage specificity

A new type of substrate was developed to verify the results obtained from the phage display analysis. Two copies of the *E. coli* thioredoxin gene were inserted in tandem into the pET21 vector for bacterial expression. In the C-terminal end a His$_6$- tag was inserted for purification on $Ni^{2+}$ IMAC columns. In the linker region, between the two thioredoxin molecules, the different substrate sequences were inserted by ligating double stranded oligonucleotides into two unique restriction sites, one BamHI and one SalI site (Fig 5A). The sequences of the individual clones were verified after cloning by sequencing of both DNA strains. The plasmids were then transformed into the *E. coli* Rosetta gami strain for protein expression (Novagen, Merck, Darmstadt, Germany). A 10 ml overnight culture of the bacteria harbouring the plasmid was diluted 10 times in LB + Amp and grown at 37 ˚C for 1–2 hours until the OD (600 nm) reached 0.5. IPTG was then added to a final concentration of 1 mM. The culture was then grown at 37˚C for an additional 3 h under vigorous shaking, after which the bacteria were pelleted by centrifugation at 3500 rpm for 12 minutes. The pellet was washed once with 25 ml PBS + 0.05% Tween 20. The pellet was then dissolved in 2 ml PBS and sonicated 6 x 30 seconds to open the cells. The lysate was centrifuged at 13000 rpm for 10 minutes and the supernatant was transferred to a new tube. Five hundred μl of Ni-NTA slurry (50:50) (Qiagen, Hilden, Germany) was added and the sample was slowly rotated for 45 min at RT. The sample was then transferred to a 2 ml column and the supernatant was allowed to slowly pass through the filter leaving the Ni-NTA beads with the bound protein in the column. The column was then washed four times with 1 ml of washing buffer (PBS + 0.05% Tween + 10 mM Imidazole + 1 M NaCl). Elution of the protein was performed by adding 150 μl elution buffer followed by five 300 μl fractions of elution buffer (PBS + 0.05% Tween 20 + 100 mM Imidazole). Each fraction was collected individually. Ten μl from each of the eluted fractions was then mixed with 1 volume of 2 x sample buffer and 1 μl β-mercapto-ethanol and then heated for 3 min at 80˚C. The samples were analysed on a SDS bis tris 4–12% pre-cast SDS-PAGE gels (Invitrogen, Carlsbad, CA, USA) and the second and third fractions that contained the most protein were pooled. The protein concentration of the combined fractions was determined by Bio-Rad DC

Protein assay (Bio-Rad Laboratories Hercules, CA USA). Approximately 60 µg of recombinant protein was added to each 120 µl cleavage reaction (in PBS). Twenty µl from this tube was removed before adding the enzyme for the 0 minute time point. The active enzyme was then added and the reaction was kept at room temperature during the entire experiment. Twenty µl samples were removed at the indicated time points (15 min, 45 min and 150 min) and stopped by addition of one volume of 2 x sample buffer. One µl β-mercapto-ethanol was then added to each sample followed by heating for 3 min at 80˚C. Twenty µl from each of these samples was then analysed on 4–12% pre-cast SDS-PAGE gels (Invitrogen, Carlsbad, CA, USA). The gels were stained over-night in colloidal Coomassie staining solution and de-stained for several hours according to previously described procedures [49].

## Supporting information

**S1 Fig. Analysis of the chromosomal loci for several additional lysozyme c related genes.** We here show the loci of three additional lysozyme c related genes, LysL1, LysL4-Like and Lyz-Like6 for a comparison to show that gene duplications primarily have occurred in the lysozyme c locus as shown in Fig 1.
(DOCX)

**S1 Table.**
(DOCX)

**S1 Raw images.**
(PDF)

## Author Contributions

**Conceptualization:** Michael Thorpe, Lars Hellman.

**Data curation:** Zhirong Fu, Srinivas Akula, Chang Qiao, Jinhye Ryu, Lars Hellman.

**Formal analysis:** Zhirong Fu, Srinivas Akula, Michael Thorpe, Lars Hellman.

**Funding acquisition:** Lars Hellman.

**Investigation:** Zhirong Fu, Srinivas Akula, Chang Qiao, Jinhye Ryu, Gurdeep Chahal.

**Project administration:** Lars Hellman.

**Resources:** Lawrence de Garavilla, Jukka Kervinen.

**Supervision:** Lars Hellman.

**Validation:** Zhirong Fu, Lars Hellman.

**Visualization:** Zhirong Fu, Lars Hellman.

**Writing – original draft:** Lars Hellman.

**Writing – review & editing:** Zhirong Fu, Gurdeep Chahal, Jukka Kervinen, Michael Thorpe.

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
