## [Decision Letter · Decision Letter 0]

10 Mar 2021

PONE-D-20-40762

Duodenases are a small subfamily of ruminant intestinal serine proteases that have undergone a remarkable diversification in cleavage specificity

PLOS ONE

Dear Dr. Hellman,

Thank you for submitting your manuscript to PLOS ONE. After careful consideration, we feel that it has merit but does not fully meet PLOS ONE’s publication criteria as it currently stands. Therefore, we invite you to submit a revised version of the manuscript that addresses the points raised during the review process.

Both reviewers make cogent criticisms, and make constructive suggestions as to how the manuscript can be improved. Please address the issues that they raise as fully as possible in your revised manuscript.

We look forward to receiving your revised manuscript.

Kind regards,

Israel Silman

Academic Editor

PLOS ONE

Journal Requirements:

2. We noticed you have some minor occurrence of overlapping text in the Results section with the following previous publications, which needs to be addressed:

- https://journals.plos.org/plosone/article?id=10.1371%2Fjournal.pone.0131720

- https://www.frontiersin.org/articles/10.3389/fimmu.2018.02387/full?

- https://journals.plos.org/plosone/article?id=10.1371/journal.pone.0195077

In your revision ensure you cite all your sources (including your own works), and quote or rephrase any duplicated text outside the methods section. Further consideration is dependent on these concerns being addressed.

'This study was supported by grants from Knut and Alice Wallenberg foundation KAW2017.0022.'

'The funders had no role in study design, data collection and analysis, decision to publish, or preparation of the manuscript.'

6. Thank you for stating the following in the Competing Interests section:

'The authors have declared that no competing interests exist.'

We note that one or more of the authors are employed by commercial companies: GDL Pharmaceutical Consulting and Contracting,  and Tosoh Bioscience LLC

a. Please provide an amended Funding Statement declaring these commercial affiliations, as well as a statement regarding the Role of Funders in your study. If the funding organization did not play a role in the study design, data collection and analysis, decision to publish, or preparation of the manuscript and only provided financial support in the form of authors' salaries and/or research materials, please review your statements relating to the author contributions, and ensure you have specifically and accurately indicated the role(s) that these authors had in your study. You can update author roles in the Author Contributions section of the online submission form.

If your commercial affiliations did play a role in your study, please state and explain this role within your updated Funding Statement.

b. Please also provide an updated Competing Interests Statement declaring these commercial affiliations along with any other relevant declarations relating to employment, consultancy, patents, products in development, or marketed products, etc.  

Reviewers' comments:

Reviewer's Responses to Questions

**Comments to the Author**

1. Is the manuscript technically sound, and do the data support the conclusions?

Reviewer #1: Partly

Reviewer #2: Yes

2. Has the statistical analysis been performed appropriately and rigorously? 

Reviewer #1: I Don't Know

Reviewer #2: Yes

3. Have the authors made all data underlying the findings in their manuscript fully available?

Reviewer #1: Yes

Reviewer #2: Yes

4. Is the manuscript presented in an intelligible fashion and written in standard English?

Reviewer #1: Yes

Reviewer #2: No

5. Review Comments to the Author

Reviewer #1: This work contains findings attractive to those interested in serine proteases, mammalian evolution, and ruminant biology. The experiments are mostly well performed but there are concerns enumerated below.

General comments

1. The lysozyme portion of the manuscript is related peripherally to the manuscript’s main subject, which is substrate preferences of duodenase-like peptidases of ruminants and pigs. Despite the large amount of text devoted to lysozyme in Abstract, Introduction and Discussion, and inclusion of lysozyme genes in two figures, lysozyme appears neither in the title nor as a key word, reflecting its secondary importance to this lengthy manuscript. Removing much of the lysozyme material would shorten and focus this work, making it easier to receive the main message.

2. The authors have used artificial substrates to identify extended peptide substrate patterns that predict potential cleavage sites in native proteins. The paper would be improved if at least one of these were followed up by actually testing the rather speculative predictions. An obvious experiment is to test bovine MCP1A versus mucin 5B. It is by no means a given that cleavage would occur in a mucin, given shielding of potential peptide cleavage sites by the abundant sugars attached to mucin-class macromolecules.

3. Given this work’s focus on sheep duodenase-related enzymes, it should include a reference to the seminal work of Pemberton, Huntley and Miller (Biochem J 321:665-670, 1997). Apparently, they were the first to show the “dual specificity” of a sheep duodenase/chymase. Their findings using purified native enzyme are highly relevant to the present work with recombinant enzymes.

Specific comments

1. The background material and discussion regarding duodenases presumes a digestive function, but in most cases the cellular origin and function of these proteases is unknown, which deserves acknowledgement. Presence in gut tissues does not preclude a primary immune function, such as those of many of these enzymes’ close relatives among serine proteases with established immune function.

2. Page 8, 1st paragraph. “Phase display” should be “Phage display”

3. There is near-complete absence of methods, statistical techniques, and references to source sequences relating to results of phylogenetic analysis depicted in Fig 3. For example, what was used in the analysis: gene sequence, cDNA, or deduced protein sequence, and how was it prepared for alignment?

4. The Fig 14 legend refers to a phylogenetic tree in panel B, but Fig 14 contains only one panel and lacks a tree. Where is panel B?

5. Fig 3 is problematic. In the “Duodenase” clade, Sheep MCP1A and Sheep MCP3-lke each appear twice, i.e., are duplicated. Apparently, something was mislabeled. In the “Cathepsin G“ clade, “Human” is misspelled. In the “Mast Cell Chymases” clade, Sheep and Bovine MCP-2 are duplicated and at least two of these must be mislabeled. “Mouse FogFX” presumably should be “Mouse CogFX”, though “CogFX” is a non-standard abbreviation and is undefined. “rMcp-5” and “mMCP-5 do not identify the mammalian source and thus are inconsistent with the labeling of every other protease in the tree (also it should be noted that standard nomenclature for these presumptive rat and mouse genes is Cma1, because they are recognized to be the orthologues of human CMA1).

7. Figure 15: “Chimpanzee” is misspelled.

8. Page 43: “Supplementary” is misspelled.

Reviewer #2: The manuscript by Fu and colleagues describes an interesting study of the expanded family of ruminant intestinal serine proteases that display functional variety. The authors have used a variety of methods to elucidate the potential functionality of these proteins based on their substrate specificity to provide further insights into these molecules. Although this is an interesting study, the manuscript is quite long, is difficult to read in parts and is written in parts more colloquially rather than scientifically. I recommend the authors edit the manuscript. I also have the following comments/questions:

1. Line numbers would have made the review easier.

2. The introduction is long and reads more like a review article rather than an introduction setting up the manuscript. Parts of the introduction is also duplicated in the discussion.

3. Introduction - avoid using 'we have' when describing the number of genes within different genomes.

4. Introduction and throughout the manuscript - be consistent when referring to numbers below 10, standard convention is to write the numbers out in full to the number ten and use the numerical format afterwards.

5. Results paragraph one - the authors describe another chymase MCP2 - is this the same MCP2 referred to in the sentences above? It is unclear as to whether this chymase was recombinantly expressed?

6. Results - be more concise in the results giving the important points, rather than describing every process, which would be more typical in a thesis rather than a manuscript.

7. Relating to figure 4 - did the authors perform a Western blot with an anti-His tag antibody to confirm removal of the his tag?

8. Relating to figure 5 - do the authors know whether substrate cleavage occurs at a preferred pH? The authors should also use this data to analyse the enzyme kinetics rather than including the curves as a figure.

9. Discussion - please include discussion points relevant to the results.

10. Methods - this section requires extensive editing to provide clear methods so that the readers can understand how the results were obtained. The authors should include methods regarding their PCR and bioinformatic analyses. The authors state that the gene sequences used for recombinant expression were obtained from Genbank - please include the accession numbers.

11. Figures - the authors have included a lot of figures, some of which should be moved to supplemental data. The majority of the figures are complex, that includes text or parts that are too small. These should be edited in accordance with PLoS one's recommendations for figures.

6. PLOS authors have the option to publish the peer review history of their article (what does this mean?). If published, this will include your full peer review and any attached files.

Reviewer #1: No

Reviewer #2: No

---

## [Author Response · Author response to Decision Letter 0]

9 Apr 2021

Dear Editor and reviewers,

We have now revised the manuscript according to the two reviewers recommendations.

All changes in the text except new references have been marked in red, except deleted text that has just been deleted.

We have also removed the ¨data not shown¨ on pages 7 and 9 as suggested by the editorial office. 

The suggested text concerning the Funding disclosure and Competing interest as suggested by the editorial office is fine with us, as shown below.

Funding Disclosure: “GDL Pharmaceutical Consulting and Contracting provided support for this study in the form of salary for LG and Tosoh Bioscience LLC provided support for this study in the form of salary for JK. The specific roles of these authors are articulated in the ‘author contributions’ section. Knut och Alice Wallenbergs Stiftelse provided support to LH (KAW2017.0022). The funders had no role in study design, data collection and analysis, decision to publish, or preparation of the manuscript.”

Competing Interests: “The authors have read the journal’s policy and the authors of this manuscript have the following competing interests: LG is a paid employee of GDL Pharmaceutical Consulting and Contracting and JK is a paid employee of Tosoh Bioscience LLC. There are no patents, products in development or marketing products to declare. This does not alter our adherence to PLOS ONE policies on sharing data and materials.”

Please add this text to the manuscript.

We have also compiled all raw figures into one PDF file name S1 raw images as suggested by the editorial office and uploaded it as supplementary material. We have tried to make it even more clear by adding headings to all lanes included in figure 4 and also a materials and methods heading in this PDF to give details on how the original images have been generated. 

Here follows the response to the reviewers comments.

Reviewer 1. 

1. We agree that the section on lysozyme is quite extensive. However, we feel that part is essential for the understanding of the importance of gene amplifications in the process of adaptation for this complex and difficult food source in cellulose. The combination of the lysozyme and the duodenases is what in our mind make the duodenases extra relevant. To shorten the lysozyme part moved figure 15 which is the second figure of lysozyme to the supplementary material. Now only one out of 14 figures describe lysozyme. The only remaining lysozyme figure is now figure 1 which describes the massive gene amplification of this locus in ruminants. To our knowledge no other locus is presently known to have experienced a similar expansion in ruminants to aid food digestion. We have also tried to reduce redundancies in the description of the lysozyme to shorten he manuscript.

2. We agree that it would have been great with a study of Mucin 5B. We have therefore spent almost half a year trying to get such data. We obtained saliva from three cows and we ordered the most well-established combination of glycosidases available on the market, the deglycosylation mix from New England Biolabs. We tested several times to deglycosaylate the salivary mucins to be able to test cleavage with the duodenases and we observed cleavage of the smaller bands not being mucin 5B. However, mucin 5B never entered the gel as one band due to major difficulties to deglycosylate mucins as also stated in the NEB prospect for the deglycosylation kit. They say that this kit is not suitable for mucins as they are notoriously difficult to deglycosylate due to the very extensive glycosylation as can be seen from the highly repetitive structure of the regions of mucin 5B containing repeated Ser and Thr residues used for attaching the O-linked carbohydrates. So we had a figure but one of our coauthors when editing the manuscript did not think it added any value to the manuscript as we were not able to show mucin 5B cleavage as the partial deglysosylatin resulted in smears barely entering the gel. So we have tried and we agree that it would be great but with the most powerful tools on the marked we have not been able to convincingly determine the extent of cleavage of mucin 5B. We will try to solve this issue and hope it can be part of a coming article.

3. We are fully aware of the great articles by Pemberton, Huntley and Miller and we have had a very productive collaboration with the Miller lab for several years on the analysis of the mucosal mast cell proteases. This article on duodenases was also part of the initial manuscript but by unknown reasons been omitted in the last revised version. We have now added a section in the introduction and results discussing this earlier duodenase study. Concerning the dual chymase tryptase specificity. Studies of purified proteases from a tissue can sometimes cause problems with specificity as these enzymes are very similar in structure, size and charge and a single band can actually be two or even three enzymes with similar properties in the same gel band why one cannot exclude that a purified band contains one enzyme with chymase activity and one with tryptase activity. Purified recombinant enzymes is therefore to prefer as then we can with 100% certainty say that the enzyme actually have dual activity. This sheep enzyme is also mainly a tryptase as the chymase activity is 160 times lower than the tryptase activity. This we show with the recombinant substrates. 

Concerning tissue distribution. The tissue distribution of the duodenases is nicely shown by the Russian lab as they show that the duodenal secretion contains exceptionally high levels of these proteases and that no expression was detected in immune cells of the intestinal samples. We have not our selves verified this, except for the PCR analysis of duodenal tissue, so we have to rely on the validity of their data but it looks very convincing with both immuno-histochemistry and immune gold staining and also measurements of protein concentrations in duodenal homogenate. This is why we doubt that these duodenases are immune proteases in the classical sense. However, we have added the information concerning immunohistochemical staining that indicate expression also in mucosal mast cells, at least during inflammatory conditions, and also several of the Miller lab references. It is however a risk of cross reactivity of the antiserum to other related proteases as mast cells are loaded with other related proteases why more detailed analysis of the presence of duodenases in mast cells is needed before we can make any conclusions about their expression also in immune cells. I admit that the cloning from abomasum and BMMCs is a strong indication for their expression in mast cells which we have added to both intro and discussion sections so the question needs further analysis which we now also state in the discussion. 

Specific comments

1. Described above comment 3.

2. Phase have been corrected.

3. Materials and methods section has been updated including PCR, chromogenic assay and phylogenetic tree sections where all the accession numbers for the amino acid sequences for the proteases included in the tree has been added. We have also added primer sequences for PCR and also the chromogenic substrates.

4. Figure 4 had initially a tree which was removed and no one of us did notice this during the numerous readings of the manuscript. So thanks for observing this. It has now been removed. 

5. Human has been corrected in the cathepsin G clade. In the genome annotation both MCP2 genes have this name, the same sheep MCP-3 like. We have again double checked and that’s the way they are annotated in the present genome sequence. Foag has been corrected. Cma1 is normally used for the primates whereas MCP-5 is the most commonly used name for the Cma1 in mouse and rat. 

6. –

7. Chimpanzee has been corrected.

8. Supplementary has been corrected. 

Reviewer 2.

The manuscript is written more colloquially: Response: In my mind one of the most difficult but also most desirably things in science is to describe a difficult phenomenon with simple words and to use a strict logic to enable as many as possible to understand and interpret the new findings presented. This is what we have tried to write the article in an easily understandable form. For example, if one reads highly respectable scientific journals like ¨Cell¨ they are highly skilled in presenting a subject correct but logic and very easy to follow. First giving the rationale why an experiment is done, followed by the setup of the experiment in simple words and then give the results in an easily understandable form. This is to me the golden standard in presenting scientific data. I know that some journals particularly in the Veterinary field often leave out the rationale behind an experiments and only give the final data without any explanation of how the data have been obtained, which in my mind make the articles almost unreadable. I do not know how we should have made more scientific by using more difficult words or leaving out important descriptions of the reasons why we do the experiment? I think that one the most important tasks for the science community is to make the new findings understandable and therefore to make the description as easy as possible to read and to understand the relevance of the new findings, and hopefully also put the data in a broader context. I must confess I do not know how we could have made this better. 

1. Sorry for the lack of numbering

2. We have looked for overlaps and tried to reduce. However, the introduction is not longer than normal and it tries to put the duodenases in the context of gene duplications in the evolution of adaptation to cellulose rich food in ruminants, which at least I think is highly interesting. These two loci complement each other in a very nice way giving a broader view of the complex task in cellulose digestion and possibly clues to the function of the duodenases. This is also why we need to describe both gene loci which make the text slightly longer than a short intro, but not longer than average. 

3. We have, have been replaced by ¨there are¨.

4. We have now removed all numbers under 10 and replaced by them with the numbers in writing. 

5. The chymase MCP2 was expressed at the same time as MCP3, the duodenase, in insect cell. This has been added to the text.

6. As described in the first part of the response we feel its important to give the rationale, the experiment and then the result, which is the most common way of doing it in most high impact journals like Cell, Nature and Science to make it easy for the reader to understand why the experiment is done and how to interpret the data. 

7. The enzyme is not active if not the His tag is removed and the difference in size is substantial why one directly can see from the gel if the His tag including the enterokinase site has been removed. As can be seen from the figure almost all of the enzymes have nearly 100% removal of the His-Enterokinase tag. It is only the pig enzyme that has some remaining uncleaved material. However, this enzyme is very active why a small remaining amount of uncleaved has little effect on the result in both phage display and 2xTrx analysis. 

8. All the studies are done at pH 7.2 which is the approximate pH of the duodenum and should thereby match the normal environment for the enzyme. These enzymes have however been shown to have higher activity at higher pH but the environment where they act are most likely in the duodenum where the pH is around 7. 

Concerning enzyme kinetics. To have any value of enzyme kinetics the kinetics should be determined for the natural substrate, which we do not yet know. Enzyme kinetics with chromogenic substrates is as far one can come from natural conditions and does therefore not add any valuable kinetic information of the enzyme. It gives a number and is therefore present in many articles on enzymes, but these numbers have little if any biological relevance as the kinetics with natural substrates can be orders of magnitude different from these surrogate values. As we have shown for many of these enzymes residues C-terminally of the cleavage site is of major importance for cleavage activity and selectivity and the chromogenic substrates have no C-terminal residues why they are highly un-physiological. Concerning the presentation of the chromogenic assay. At least to me, and I think most readers, it is easier to see the difference in activity by looking at the graph than having a numeric value of kinetics in the text. The chromogenic substrate assay is also just an initial rough test of activity to guide the interpretation of the phage display and the recombinant substrates which is much more accurate in determining the actual biological activity of the enzyme as they both contain both N and C terminal residues of the cleavage site, although in a linear form.

9. Sorry I do not understand we have a number of points in the discussion related to the results of the article??? The first page and the entire last one and a half page of the discussion focuses entirely on the duodenases. Th rest of the discussion is devoted to other loci involved in the complex question of digestion of cellulose to put the duodenases in a larger biological context. 

10. We have now added sections in the Materials and methods section describing PCR, Chromogenic substrate assay and bioinformatic analyses and also all accessions numbers for the protein sequences used for the tree and the primer sequences used for the PCR, and also all the chromogenic substrates. 

11. We have now removed figure 15 to the supplementary section. To enlarge the text in the figures have been difficult as the only two figures where the text is small is in the gene figure where the names cannot be bigger as they then go into each other. In the chromogenic substrate assay the substrates have been added to the materials and methods section if they are difficult to read in the figure.

We have removed the ¨data not shown¨ and given the reference to that data that is already published. We also include all uncropped gels for this manuscript for the editorial office to go through. 

We hope these changes now make the manuscript suitable for publishing in PlosOne.

Sincerely

Lars Hellman PhD Professor.

Uppsala University

Dept .of Cell and Molecular Biology

BMC, Box 596, SE-751 24 Uppsala, SWEDEN

---

## [Decision Letter · Decision Letter 1]

19 May 2021

Duodenases are a small subfamily of ruminant intestinal serine proteases that have undergone a remarkable diversification in cleavage specificity

PONE-D-20-40762R1

Dear Dr. Hellman,

We’re pleased to inform you that your manuscript has been judged scientifically suitable for publication and will be formally accepted for publication once it meets all outstanding technical requirements. Please note that Reviewer 1 has suggested that 'lysozyme' be added to the list of Key Words.

Kind regards,

Israel Silman

Academic Editor

PLOS ONE

Additional Editor Comments (optional):

Reviewers' comments:

Reviewer's Responses to Questions

**Comments to the Author**

1. If the authors have adequately addressed your comments raised in a previous round of review and you feel that this manuscript is now acceptable for publication, you may indicate that here to bypass the “Comments to the Author” section, enter your conflict of interest statement in the “Confidential to Editor” section, and submit your "Accept" recommendation.

Reviewer #1: All comments have been addressed

Reviewer #2: All comments have been addressed

2. Is the manuscript technically sound, and do the data support the conclusions?

Reviewer #1: Yes

Reviewer #2: Yes

3. Has the statistical analysis been performed appropriately and rigorously? 

Reviewer #1: Yes

Reviewer #2: Yes

4. Have the authors made all data underlying the findings in their manuscript fully available?

Reviewer #1: Yes

Reviewer #2: Yes

5. Is the manuscript presented in an intelligible fashion and written in standard English?

Reviewer #1: Yes

Reviewer #2: Yes

6. Review Comments to the Author

Reviewer #1: I suggest adding lysozyme to the list of key words. Otherwise the authors have addressed my concerns adequately.

Reviewer #2: (No Response)

7. PLOS authors have the option to publish the peer review history of their article (what does this mean?). If published, this will include your full peer review and any attached files.

Reviewer #1: No

Reviewer #2: No

---

## [Editor Report · Acceptance letter]

21 May 2021

PONE-D-20-40762R1 

Duodenases are a small subfamily of ruminant intestinal serine proteases that have undergone a remarkable diversification in cleavage specificity 

Dear Dr. Hellman:

I'm pleased to inform you that your manuscript has been deemed suitable for publication in PLOS ONE. Congratulations! Your manuscript is now with our production department. 

Kind regards, 

on behalf of

Prof. Israel Silman 

Academic Editor

PLOS ONE